# CoT-Evo: Evolutionary Distillation of Chain-of-Thought for Scientific Reasoning

**Kehua Feng**[1,2]**, Keyan Ding**[1,2]*, **Zhihui Zhu**[1]**, Lei Liang**[3]**, Qiang Zhang**[1,2]**, Huajun Chen**[1,2]
[1]Zhejiang University   [2]ZJU-Hangzhou Global Scientific and Technological Innovation Center
[3]AntGroup
{kehuafeng, dingkeyan}@zju.edu.cn

## Abstract

While chain-of-thought (CoT) distillation from advanced large language models (LLMs) has proven effective in general reasoning tasks, it struggles in scientific domains where even advanced models often produce incorrect or superficial reasoning due to high complexity and specialized knowledge requirements. Directly distilling from such flawed outputs results in low-quality training data and limits the performance of smaller student models. To overcome this, we propose CoT-Evo, an evolutionary CoT distillation framework. It begins by constructing a diverse pool of reasoning trajectories from multiple LLM thinkers, enriches them with automatically retrieved domain knowledge, and iteratively refines the trajectories using novelty-driven selection, reflective recombination and mutation. The refinement is guided by a fitness function that evaluates answer correctness, coherence, and effective knowledge utilization. This results in a high-quality CoT dataset tailored for scientific reasoning. We employ this evolved dataset to fine-tune a compact model, which achieves state-of-the-art performance on scientific reasoning benchmarks. Our work establishes a scalable approach to synthesizing high-fidelity scientific reasoning data from diverse and fallible LLMs. Code is available at https://github.com/weiji-Feng/MAD-Eval.

## 1 Introduction

Recent advances in reasoning large language models (LLMs), such as DeepSeek-R1 (Guo et al., 2025) and OpenAI-o1/o3 (Jaech et al., 2024), have demonstrated that leveraging long and structured chains of thought (CoTs) leads to remarkable improvements in complex reasoning tasks. CoT distillation from advanced teacher models has proven effective in general domains (Ye et al., 2025; Hu et al., 2025). However, when applied to scientific domains, even the strongest LLMs frequently generate erroneous or superficial reasoning paths due to the increasing complexity and specialization of scientific tasks (Liu et al., 2025; Li et al., 2025). This raises a critical need for more fine-grained CoT distillation approaches tailored to the unique requirements of scientific reasoning.

Existing work has made significant progress in optimizing CoT distillation. Some approaches focus on enhancing intra-chain quality in single-teacher settings by compressing reasoning token length (Wang et al., 2025; Lu et al., 2025) or identifying intermediate error states (Luo et al., 2025). Others employ multi-teacher to aggregate diverse reasoning paths and select the most suitable CoT for each sample (Zhu et al., 2024; Xu et al., 2025). While these methods have enriched the landscape of CoT distillation, they overlook two critical aspects that are particularly consequential in scientific reasoning tasks: 1) *single-teacher optimization* may introduce bias (Xu et al., 2025), and merely pruning redundant or erroneous steps does not ensure accurate knowledge usage in the core thought; 2) *multi-teacher framework* increases diversity but lacks the flexibility to refine the internal logic of CoTs at a fine-grained level.

To address these limitations, we propose CoT-Evo, an evolutionary CoT distillation framework. Distinct from prior work that merely selects or transfers a single promising reasoning path per sample, CoT-Evo performs *intra-chain* aggregation, dynamically integrating thoughts from multiple

---

*Corresponding author.

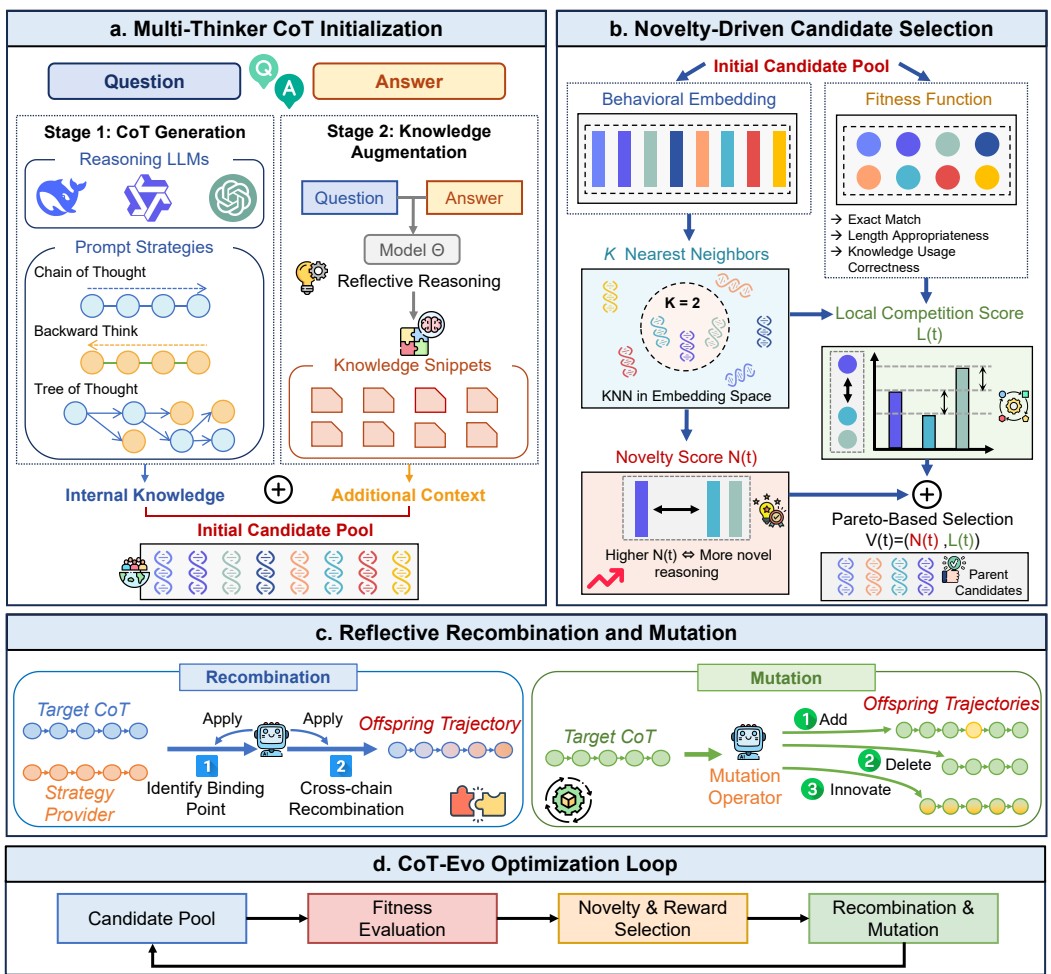

Figure 1: Overview of CoT-Evo. (a) *Multi-thinker CoT initialization* constructs a diverse pool of candidates. (b) *Novelty-driven candidate selection* evaluates them via a composite *fitness function* and retains promising trajectories. (c) *Reflective recombination and mutation* generate new offspring CoTs through targeted operations. (d) These modules form the iterative CoT-Evo *optimization loop*, which evolves compact, accurate, and domain-reliable CoTs for downstream training.

CoTs to synthesize a single, high-quality chain. Specifically, CoT-Evo begins by generating a diverse candidate pool of CoTs using multiple LLMs and prompting strategies, optionally augmented with external knowledge. Each candidate is then scored by a fitness function measuring answer correctness, reasoning length appropriateness, and knowledge usage accuracy. Rather than greedily picking the top candidates, CoT-Evo employs a *novelty-driven selection* mechanism that jointly rewards quality and behavioral diversity. Finally, reflective recombination and mutation operators integrate or revise reasoning steps across candidates to produce improved offspring chains. Iterating this evaluation–selection–variation–update loop yields a compact yet high-fidelity set of evolved CoTs for downstream training.

Our **contributions** can be summarized as follows:

- We introduce the first intra-chain multi-trajectory aggregation framework for CoT distillation, enabling fine-grained integration of reasoning steps within a single chain.
- We propose an evolutionary CoT distillation approach that integrates novelty-driven selection, reflective recombination, and mutation to iteratively refine reasoning trajectories.
- We construct an evolved CoT dataset and demonstrate that our method significantly improves the performance of student models on scientific reasoning tasks.

## 2 RELATED WORK

**Long CoT Distillation**   Long CoTs have been shown to significantly enhance LLMs' reasoning ability, as evidenced by models like DeepSeek-R1 (Guo et al., 2025) and OpenAI-o1/o3 (Jaech et al., 2024). Distilling such trajectories from strong teacher models into smaller ones has become a common strategy to boost reasoning efficiency under limited computation (Ye et al., 2025; Hu et al., 2025). To improve quality, prior work has explored intra-chain optimization, *e.g.*, compressing reasoning token length or pruning erroneous steps (Wang et al., 2025; Luo et al., 2025), and inter-chain (*i.e.*, multi-teacher) strategies, *e.g.*, aggregating multiple reasoning paths from different teachers or prompting paradigms (Chain-of-Thought (Wei et al., 2022), Tree-of-Thought (Yao et al., 2023), or Program-of-Thought (Chen et al., 2022)) (Zhu et al., 2024; Xu et al., 2025; Lei et al., 2025). Despite these advances, existing methods remain limited in scientific domains, where ensuring both factual accuracy and logical rigor is critical. To this end, we introduce COT-EVO, which applies an evolutionary, fine-grained intra-chain aggregation strategy to combine and refine reasoning steps across CoTs, yielding more reliable and domain-accurate scientific reasoning trajectories.

**Reasoning in Scientific Domains**   With the rapid progress of LLMs in mathematics and code, recent efforts have turned to evaluating their capabilities in scientific domains. Benchmarking efforts have shifted focus from high school-level exams such as MMLU (Hendrycks et al., 2020), CEval (Huang et al., 2023), and AGIEval (Zhong et al., 2023), to more advanced scientific tests including GPQA (Rein et al., 2024), SciEval (Sun et al., 2024), and SciKnowEval (Feng et al., 2024), and further to domain-intensive tasks such as BioMaze (Zhao et al., 2025a), BioProBench (Liu et al., 2025), and ChemCoTBench (Li et al., 2025), which emphasize reasoning over memorization, covering areas from biological protocols to molecular design. Moreover, developing specialized scientific LLMs to tackle complex downstream tasks has emerged as a promising direction (Zhao et al., 2025b; Bai et al., 2025). In this paper, we aim to provide a generalizable CoT distillation framework that generates high-quality reasoning trajectories adaptable to diverse scientific applications.

## 3 COT-EVO: EVOLUTIONARY COT DISTILLATION

In this section, we present COT-EVO, a novel evolutionary framework tailored for CoT distillation in scientific reasoning tasks. Let $\mathcal{D}_{\text{ori}} = \{(x_i, y_i)\}_{i=1}^N$ denote the original training dataset without CoT. For each $x_i$, the goal of COT-EVO is to generate high-fidelity, compact CoTs for downstream training by leveraging a diverse *candidate pool* $\mathcal{P} = \{t_1, t_2, \ldots, t_n\}$ produced from a collection of LLM thinkers $\mathcal{L}$. Through iterative refinement guided by carefully crafted reward signals, COT-EVO evolves these candidates into more accurate and domain-reliable reasoning chains.

Following the principle of genetic algorithm, COT-EVO consists of four core modules: 1) multi-thinker CoT initialization (Section 3.1), 2) novelty-driven candidate selection (Section 3.3), 3) reflective CoT recombination and mutation (Section 3.4), and 4) fitness function definition (Section 3.2). We summarize the optimization loop of COT-EVO in Section 3.5. Figure 1 provides an overview of COT-EVO, and the full COT-EVO algorithm is formalized in Algorithm 1.

### 3.1 MULTI-THINKER COT INITIALIZATION

A key to the success of evolutionary algorithms lies in constructing an initial *candidate pool* $\mathcal{P}$ that is both diverse and promising. To this end, we employ a two-stage approach.

**CoT Generation**   We first assemble a collection of LLM thinkers $\mathcal{L} = \{l_1, l_2, \ldots, l_m\}$, where each $l_i$ may represent: 1) reasoning-based LLMs from different model families or scales, *e.g.*, DeepSeek-R1 (Guo et al., 2025), Qwen3-235B-A22B (Yang et al., 2025), and Qwen3-32B; or 2) instruction-tuned LLMs with varying prompting strategies, *e.g.*, Tree-of-Thought (Yao et al., 2023), Chain-of-Thought (Wei et al., 2022), and backward reasoning. Formally, for query $x \in \mathcal{D}_{\text{ori}}$, each trajectory $t_i \in \mathcal{P}^G$ is generated by

$$t_i = l_i(x), \quad i = 1, 2, \ldots, m. \tag{1}$$

**Knowledge Augmentation**   Considering the high specialization of scientific domains, a model's internal knowledge alone is often insufficient. Therefore, we collect additional knowledge required for

each query $x$ using a prompt-based automated approach. Specifically, for each QA pair $(x, y) \in \mathcal{D}_{\text{ori}}$, we prompt an advanced proprietary model $\Theta$ to perform reflective reasoning from $y$ to identify the knowledge necessary for solving $x$, and then abstract it into general, context-independent snippets $\mathcal{K}_x$. We then randomly select a subset of thinkers, and provide each chosen thinker $l_j$ with $\mathcal{K}_x$ as additional context, resulting in a knowledge-enhanced CoT $t_j \in \mathcal{P}^K$ represented as

$$t_j = l_j(x, \mathcal{K}_x), \quad j = 1, 2, \ldots, n - m. \tag{2}$$

This approach results in an initial candidate pool $\mathcal{P} = \mathcal{P}^G \cup \mathcal{P}^K$ that maximizes both cognitive difficulty (Cai et al., 2025) and strategic diversity, simulating a broad coverage of the solution space.

## 3.2 Fitness Function

We evaluate each candidate trajectory $t$ using a composite fitness score that captures 1) answer correctness, 2) reasoning-length appropriateness, and 3) correctness of knowledge usage.

**Exact Match**    We use a task-specific external script to check whether the candidate's final answer exactly matches the ground truth. The exact match score is binary:

$$s_{\text{EM}} = \begin{cases} 1 & \text{if exact match,} \\ 0 & \text{otherwise.} \end{cases} \tag{3}$$

**Length Appropriateness**    We compute token lengths from scientific reasoning datasets (*e.g.*, Llama-Nemotron-Science (Bercovich et al., 2025)) and set the 15% and 85% percentiles as lower and upper bounds, since responses shorter than the 15% threshold often miss key knowledge, while those beyond the 85% threshold tend to be verbose. Thus, the length appropriateness score is defined as

$$s_{\text{LEN}} = \begin{cases} 0.0 & \text{if len}(t) < 15\% \text{ percentile,} \\ 0.5 & \text{if len}(t) > 85\% \text{ percentile,} \\ 1.0 & \text{otherwise.} \end{cases} \tag{4}$$

This encourages concise reasoning while avoiding under-explained outputs.

**Knowledge Usage Correctness**    We employ an LLM-as-a-Judge (Zheng et al., 2023) evaluator to assess the accuracy of the knowledge applied in the core thought of $t$. Given the reference knowledge $\mathcal{K}_x$ and the CoT $t$, the judge assigns a categorical score

$$s_{\text{KNOW}} = \text{Judge}(\mathcal{K}_x, t), \quad s_{\text{KNOW}} \in \{1, 2, 3, 4, 5\}. \tag{5}$$

The final fitness function of $t$ combines the three components with weighted contributions:

$$\mathcal{R}(t) = s_{\text{EM}} + \lambda_1 s_{\text{LEN}} + \lambda_2 s_{\text{KNOW}}, \tag{6}$$

where $\lambda_1$ and $\lambda_2$ control the relative importance of length and knowledge usage.

## 3.3 Novelty-Driven Candidate Selection

While the initial pool $\mathcal{P}$ obtained in Section 3.1 offers substantial diversity in reasoning trajectories, selecting candidates solely based on the highest fitness scores can lead to premature convergence toward a narrow set of CoTs. To mitigate this, CoT-Evo employs a *novelty-driven selection* mechanism, inspired by Novelty Search with Local Competition (NSLC) (Lehman & Stanley, 2011), which simultaneously promotes two complementary objectives: (i) encouraging distinct reasoning patterns, and (ii) rewarding local quality improvements.

**Behavioral Embedding**    We first embed each CoT $t \in \mathcal{P}$ into a $d$-dimensional *behavioral space* through a mapping $b : \mathcal{P} \to \mathbb{R}^d$ capturing its structural or semantic reasoning features:

$$\mathbf{z}_t = b(t) \in \mathbb{R}^d. \tag{7}$$

In practice, we instantiate $b(\cdot)$ using *Qwen3-Embedding-8B* (Zhang et al., 2025).

**Novelty Score**  To quantify distinctiveness, we compute trajectory $t$'s average distance to its $k$ nearest neighbors $\mathcal{N}_k(t)$ in behavioral space:

$$N(t) = \frac{1}{k} \sum_{t' \in \mathcal{N}_k(t)} \left\| \mathbf{z}_t - \mathbf{z}_{t'} \right\|_2. \tag{8}$$

Larger $N(t)$ values correspond to more differentiated reasoning styles.

**Local Competition Score**  Given a fitness function $\mathcal{R}(t)$ (defined in Section 3.2) that assesses the correctness and quality of a trajectory $t$, we evaluate $t$'s relative improvement over its neighbors:

$$L(t) = \frac{1}{k} \sum_{t' \in \mathcal{N}_k(t)} \left( \mathcal{R}(t) - \mathcal{R}(t') \right)_+, \quad (a)_+ = \max(a, 0). \tag{9}$$

Here, $L(t)$ rewards local superiority within the same behavioral region.

**Pareto-Based Selection**  We treat each $(N(t), L(t))$ pair as a bi-objective performance vector $\mathbf{g}(t) = (N(t), L(t))$. Define a (strict) Pareto-dominance relation $\mathbf{g}(t') \succ \mathbf{g}(t)$, if $N(t') \geq N(t) \wedge L(t') \geq L(t)$ with at least one strict inequality. The Pareto front is then

$$\mathcal{F}_t = \{ t \in \mathcal{P} \mid \nexists t' \in \mathcal{P} \text{ s.t. } \mathbf{g}(t') \succ \mathbf{g}(t) \}. \tag{10}$$

From $\mathcal{F}_t$, parent candidates are sampled with probability

$$p(t) = \frac{L(t) + \varepsilon}{\sum\limits_{t' \in \mathcal{F}_t} \left( L(t') + \varepsilon \right)}, \quad \varepsilon > 0, \tag{11}$$

where the parameter $\varepsilon$ ensures numerical stability and slightly favors trajectories with higher local performance while maintaining diversity.

## 3.4 REFLECTIVE CoT RECOMBINATION AND MUTATION

To further evolve reasoning trajectories, CoT-EVO adopts a *reflective* approach for recombination and mutation, enabling models to incorporate useful strategies from peers or revise their own reasoning patterns.

**CoT Recombination.**  Recombination is triggered only if $t_o$ yields an incorrect final answer (*i.e.*, $s_{\text{EM}}(t_o) = 0$). Unlike standard genetic algorithms, our crossover operates on a single *target* CoT $t_o$ with guidance from a *strategy provider* $t_p$. Let $\mathcal{C}_r$ denote the recombiner (defaulting to $l_o \in \mathcal{L}$ that produced $t_o$).

*Step 1: Identify binding point.* We decompose $t_p$ into distinct *thoughts* using transition keywords (Chai et al., 2025) (see Appendix A6 for details), then apply $\mathcal{C}_r$ to select the endpoint of the last reasonable thought as the binding point $\mathcal{B}$:

$$\mathcal{B} = \mathcal{C}_r(t_p, \text{transition keywords}). \tag{12}$$

*Step 2: Cross-chain recombination.* We employ $\mathcal{C}_r$ to extract unique steps and knowledge from $t_p$ absent in $t_o$, denoted by $\mathcal{I}$, and let $\mathcal{C}_r$ generate a new CoT $t'$ conditioned on the prefix $t_o[:\mathcal{B}]$ and $\mathcal{I}$:

$$t' = t_o[:\mathcal{B}] + \mathcal{C}_r\big(t_o[:\mathcal{B}], \mathcal{I}\big). \tag{13}$$

Owing to the limited instruction-following capability of currently reasoning models, if $\mathcal{C}_r$ is a reasoning model, we explicitly insert guiding text immediately after $\mathcal{C}_r$'s initial "`<think>`" token, including: 1) "I can receive externally provided information through "`<info></info>`", and 2) "Upon receiving information, I should switch to a new thought and verify and use that information."

**Reflective Mutation.**  Mutation aims to alter $t_o$'s strategy, logic, or expression. Given $t_o$ and mutation operator $\mathcal{M}_u$ (defaulting to $l_o$), we generate new variants via three modes:

*(1) Additive mutation*: enrich logical detail, explanations, and domain knowledge:

$$t'_{(a)} = \mathcal{M}_u(t_o, \text{Add}); \tag{14}$$

*(2) Deletive mutation*: prune redundancy, unproductive exploration, or extraneous knowledge:

$$t'_{(d)} = \mathcal{M}_u(t_o, \text{Delete});$$
(15)

*(3) Innovative mutation:* diagnoses erroneous logic in $t_o$ using the correct answer $y$, then generates a new trajectory that attempts to avoid the identified mistakes:

$$t'_{(c)} = \mathcal{M}_u\big(\mathcal{M}_u(t_o, \text{Innovate}), \text{Delete}\big).$$
(16)

## 3.5 CoT-Evo Optimization Loop

Putting the above modules together, CoT-Evo proceeds in an iterative evolutionary loop that progressively refines the candidate pool, which is similar to the standard genetic algorithm Holland (1992). At each generation, the current pool $\mathcal{P}$ produced by the multi-thinker initialization (Section 3.1) is first evaluated by the fitness function $\mathcal{R}(t)$ (Section 3.2) to obtain the quality scores of all trajectories. Rather than greedily selecting only the top-scoring CoTs, we apply the novelty-driven selection mechanism (Section 3.3) to choose a set of parent trajectories that are both diverse in reasoning behavior and locally competitive. These parents are then passed to the reflective recombination and mutation operators (Section 3.4) to generate improved offspring trajectories, which inherit useful strategies or undergo targeted revisions of their reasoning steps. The resulting offspring are merged with the previous pool, with the lowest-fitness trajectories removed to maintain the population size $n_{\text{pop}}$, forming a new candidate set for the next iteration of the evaluation–selection–variation cycle. This iterative process continues until convergence or a predefined budget $B$, yielding an evolved CoT dataset $\mathcal{D}_{\text{evo}} = \{(x_i, t_i^\star, y_i)\}_{i=1}^N$ with $t_i^\star = \arg\max_{t \in \mathcal{P}_{x_i}} \mathcal{R}(t)$ for downstream model training.

## 4 Experiments

### 4.1 Implementation of CoT-Evo

Our pool of LLM thinkers includes Qwen3-32B-think (Yang et al., 2025), DeepSeek-R1 (Guo et al., 2025), and Qwen3-235B-A22B-think (Yang et al., 2025), together with three prompting strategies based on Llama4-Scout-17B-16E (Meta, 2025): *backward reasoning* (Chen et al., 2024), *Chain-of-Thought* (Wei et al., 2022), and *Tree-of-Thought* (Yao et al., 2023). For knowledge augmentation in Section 3.1, we employ GPT-5-mini to generate the knowledge snippets $\mathcal{K}_x$. In deploying the fitness function, we set $\lambda_1 = 0.3$ and $\lambda_2 = 0.1$. The optimization loop of CoT-Evo uses a fixed population size of $n_{\text{pop}} = 6$ per epoch with a computation budget of $B = 5$ epochs. During novelty-driven candidate selection, we set the $k$-nearest neighbors parameter to $k = 2$, and from the resulting Pareto front $\mathcal{F}_t$, parent trajectories are randomly sampled (three per iteration by default) for recombination and mutation. Consequently, each query can theoretically yield up to $6 + 3 \times 5 = 21$ distinct CoTs.

We apply CoT-Evo to three popular open-source LLMs: *Qwen3-8B* (Yang et al., 2025), *Qwen2.5-7B-Instruct* (Team, 2024), and *Llama3.1-8B-Instruct* (Grattafiori et al., 2024).

### 4.2 Experimental Setup

#### 4.2.1 Datasets

To comprehensively evaluate CoT-Evo, we focus on two representative scientific reasoning datasets, ChemCoTDataset and BioProBench, that capture the challenges of molecular editing and optimization, and experimental protocol design.

**ChemCoTDataset** (Li et al., 2025): This dataset contains 14K samples tailored for chemical reasoning. Complex tasks (*e.g.*, molecular design) are decomposed into modular, verifiable operations (*e.g.*, substructure addition, deletion, or replacement), ensuring that the reasoning process is transparent and step-wise. It encompasses four tasks, including molecular understanding (Und.), molecular editing (Edit), reaction prediction (Reaction), and molecular optimization (Opt.). We use ChemCoTDataset for training data construction and evaluate on its associated benchmark, ChemCoTBench.

**BioProBench** (Liu et al., 2025): This benchmark assesses LLMs' ability to reason over biological experimental protocols. It transforms 27K real-world protocols into 306K training samples and 3.6K test samples covering protocol QA (PQA), step ordering (ORD), and error correction (ERR) tasks.

Table 1: Performance comparison of CoT-Evo against baseline distillation methods and original LLM teachers across BioProBench and ChemCoTBench. Best results are marked in **bold**, and second-best results are underlined.

| Method | BioProBench | | | | ChemCoTBench | | | | | |
|---|---|---|---|---|---|---|---|---|---|---|
| | PQA | ORD | ERR | | Und. | | | Edit | Reaction | Opt. |
| | Acc. ↑ | EM ↑ | Acc. ↑ | F1 ↑ | MAE↓ | TMS↑ | Acc.↑ | Acc.↑ | SR↑ | FTS↑ |
| *LLM Teachers* | | | | | | | | | | |
| DeepSeek-R1 | **0.683** | 0.448 | 0.625 | 0.600 | **0.560** | **0.314** | 0.680 | **0.783** | **0.576** | **0.756** |
| Qwen3-32B-think | 0.628 | 0.434 | 0.629 | 0.510 | 0.575 | 0.225 | 0.691 | 0.472 | 0.406 | 0.454 |
| Qwen3-235B-A22B-think | 0.669 | **0.625** | 0.600 | **0.683** | 0.565 | 0.227 | **0.880** | 0.689 | 0.473 | 0.658 |
| Llama4-Scout-Backward | 0.595 | 0.202 | 0.625 | 0.480 | 0.850 | 0.145 | 0.655 | 0.650 | 0.402 | 0.706 |
| Llama4-Scout-CoT | 0.588 | 0.208 | 0.600 | 0.392 | 0.680 | 0.187 | 0.625 | 0.650 | 0.364 | 0.675 |
| Llama4-Scout-ToT | 0.593 | 0.181 | **0.626** | 0.539 | 0.805 | 0.172 | 0.610 | 0.589 | 0.340 | 0.634 |
| *LLM Students* | | | | | | | | | | |
| *Llama3.1-8B-Inst* | 0.468 | 0.077 | 0.580 | 0.452 | 1.860 | 0.143 | 0.615 | 0.261 | 0.147 | 0.173 |
| + Single Teacher | 0.431 | 0.341 | 0.586 | 0.667 | 0.707 | **0.260** | 0.495 | 0.640 | 0.269 | 0.527 |
| + Multi Teacher | 0.474 | 0.353 | 0.616 | 0.665 | 0.530 | 0.247 | 0.573 | 0.549 | 0.208 | 0.504 |
| + Best-of-K | 0.508 | 0.329 | 0.527 | 0.609 | 0.461 | 0.226 | 0.599 | 0.600 | 0.294 | 0.534 |
| + Retro-Search | 0.488 | 0.121 | 0.623 | 0.635 | 0.533 | 0.247 | 0.613 | 0.492 | 0.274 | 0.456 |
| + TwT | 0.499 | 0.397 | 0.570 | 0.459 | 0.607 | 0.238 | 0.597 | 0.506 | 0.278 | 0.433 |
| + CoT-Evo | **0.512** | **0.419** | **0.657** | **0.698** | **0.375** | 0.250 | **0.639** | **0.707** | **0.340** | **0.597** |
| *Qwen2.5-7B-Inst* | 0.519 | 0.227 | 0.564 | 0.398 | 0.620 | 0.086 | 0.575 | 0.178 | 0.185 | 0.304 |
| + Single Teacher | 0.518 | 0.354 | 0.593 | 0.649 | 0.929 | 0.162 | 0.533 | 0.445 | 0.186 | 0.538 |
| + Multi Teacher | 0.505 | 0.370 | 0.608 | 0.621 | 0.536 | 0.246 | 0.573 | 0.548 | 0.270 | 0.516 |
| + Best-of-K | 0.538 | 0.339 | 0.619 | 0.570 | **0.440** | 0.116 | 0.487 | 0.445 | 0.294 | 0.460 |
| + Retro-Search | 0.519 | 0.178 | 0.613 | 0.596 | 0.508 | 0.187 | 0.562 | 0.450 | 0.248 | 0.417 |
| + TwT | **0.558** | 0.358 | 0.589 | 0.438 | 0.752 | 0.132 | 0.632 | 0.442 | 0.285 | 0.448 |
| + CoT-Evo | 0.551 | **0.448** | **0.675** | **0.671** | 0.497 | **0.306** | **0.664** | **0.602** | **0.332** | **0.577** |
| *Qwen3-8B-think* | 0.602 | 0.371 | 0.590 | 0.419 | 0.440 | 0.139 | 0.612 | 0.422 | 0.278 | 0.352 |
| + Single Teacher | 0.601 | 0.368 | 0.616 | 0.534 | 0.405 | 0.268 | 0.583 | **0.629** | 0.375 | 0.599 |
| + Multi Teacher | 0.603 | 0.434 | 0.637 | 0.616 | 0.395 | 0.250 | 0.647 | 0.623 | **0.455** | 0.424 |
| + Best-of-K | 0.603 | 0.369 | 0.608 | 0.457 | 0.395 | 0.173 | 0.651 | 0.600 | 0.365 | 0.516 |
| + Retro-Search | 0.583 | 0.279 | 0.587 | 0.552 | 0.540 | 0.163 | 0.601 | 0.303 | 0.367 | 0.528 |
| + TwT | 0.475 | 0.401 | 0.563 | 0.466 | 0.865 | 0.067 | 0.540 | 0.281 | 0.319 | 0.300 |
| + CoT-Evo | **0.649** | **0.544** | **0.645** | **0.677** | **0.351** | **0.358** | **0.674** | 0.625 | 0.437 | **0.629** |

For computational efficiency, we use a 20K subset of the training QA pairs for CoT construction while retaining the full test set for evaluation.

### 4.2.2 BASELINES

For a comprehensive and fair comparison, we compare CoT-Evo with the following baselines:

- **LLM Thinkers (Teachers).** We also directly evaluate the six LLM thinkers used in our initialization stage, including DeepSeek-R1, Qwen3-235B-A22B, Qwen3-32B, and Llama4-Scout-17B-16E with three prompting strategies. These serve as the upper bound references for evaluating the distillation quality of CoT-Evo.

- **Base Models.** We directly evaluate the original instruct or reasoning LLMs without additional fine-tuning, serving as a reference to measure the overall gain of CoT-Evo.

- **Single Teacher (ST).** For each LLM thinker, we apply the fitness function (Section 3.2) to filter its CoTs and retain only those with exact-match correct answers ($s_{\text{EM}} = 1$). We repeat this process for all thinkers and report the performance of the best individual teacher.

- **Multi Teacher (MT).** For each query, we aggregate CoTs from all thinkers and directly select the correct trajectory with the highest fitness score. This representative CoT is used for training, thereby capturing diversity across teachers. If no correct CoT exists, the query is discarded.

- **Best-of-K (BoK).** For each LLM thinker, we sample $K = 21$ trajectories (matching the maximum number produced by CoT-Evo for a single query), apply rejection sampling based on the fitness function, and select the correct CoT with the highest score.

- **TwT.** TwT (Xu et al., 2025) is a recent multi-teacher distillation framework that uses *Dual-Criteria Rejection Sampling* to build a high-quality and diverse CoT corpus from multiple teacher LLMs, and then progressively distills these reasoning patterns into the student model. We implement TwT with the same set of LLM thinkers as teachers as in CoT-Evo.

- **Retro-Search (Retro).** Retro-Search (Lu et al., 2025) is an MCTS-inspired search algorithm that retrospectively explores alternative continuations of a teacher's reasoning trajectory to mitigate under-thinking and over-thinking while preserving the correct final answer. We apply Retro-Search separately to each LLM thinker and report the best student performance.

### 4.2.3 EVALUATION METRICS

For **BioProBench**, we evaluate the protocol QA (PQA) task using Accuracy (Acc), the step ordering (ORD) task using Exact Match (EM), and the error correction (ERR) task using both Acc and F1 score. For **ChemCoTBench**, we adopt task-specific evaluation metrics. In molecular understanding (Und.), we apply mean absolute error (MAE) for functional group counting, Tanimoto molecule similarity (TMS) for Murcko scaffold extraction, and accuracy for the remaining subtasks. Molecular editing (Edit) is evaluated with accuracy, reaction prediction (Reaction) is assessed by fingerprint similarity (FTS) with reference molecules, and molecular optimization (Opt.) is measured by success rate (SR), indicating the generated molecules that achieve the desired property improvement.

### 4.3 MAIN RESULTS

**CoT-Evo delivers consistent performance gains over distillation baselines.** Table 1 reports the performance of CoT-Evo compared with existing distillation methods. Our approach consistently outperforms both single-teacher (ST) and multi-teacher (MT) baselines across the two benchmarks. On BioProBench, CoT-Evo yields relative gains of 12.6% over ST and 8.4% over MT, while on ChemCoTBench the improvements reach 27.0% and 19.3%, respectively. CoT-Evo further surpasses recent strong process-level distillation baselines, including Retro-Search (Retro) and TwT. Notably, on several tasks, CoT-Evo even rivals the performance of the LLM teachers themselves. Such gains stem from CoT-Evo not only filtering out erroneous or redundant reasoning from teachers but also integrating complementary reasoning strategies across multiple thinkers, yielding distilled CoTs that are often more accurate and task-aligned than the raw outputs of the teacher LLMs. These results highlight the overall advantage of evolutionary CoT distillation in producing stronger student models.

**The performance gains of CoT-Evo go beyond the effect of increased sampling.** As described in Section 4.1, CoT-Evo can theoretically generate up to 21 candidate CoTs per query. To examine whether its superior performance merely stems from this enlarged sampling budget, we compare CoT-Evo with the Best-of-K (BoK) baseline, where each teacher independently samples $K = 21$ CoTs and selects the best one according to the fitness function. Results in Table 1 demonstrate that CoT-Evo substantially outperforms BoK, confirming that the improvements arise from higher-quality CoT evolution rather than brute-force data scaling.

**CoT-Evo improves both data utilization and reasoning quality.** To validate the source of these improvements, we analyze how CoT-Evo compares with baseline approaches in terms of usable data and CoT quality. For *data utilization*, we measure the proportion of distilled trajectories that are correct and usable. For *reasoning quality*, we employ GPT-5 as a judge to score CoTs across five dimensions: knowledge diversity, knowledge accuracy, reasoning diversity, logical coherence, and step redundancy. In addition, we conduct

Table 2: Comparison of data utilization (Pass), reasoning quality (Quality) and win rate (WR) across distillation baselines and CoT-Evo.

| Method | BioProBench | | | ChemCoTDataset | | |
|---|---|---|---|---|---|---|
| | Pass | Quality | WR | Pass | Quality | WR |
| ST | 0.375 | 6.741 | 0.329 | 0.349 | 5.702 | 0.319 |
| MT | 0.536 | 6.776 | 0.374 | 0.524 | 5.935 | 0.390 |
| BoK | 0.498 | 6.763 | 0.353 | 0.389 | 5.571 | 0.335 |
| Retro | 0.446 | 7.024 | 0.362 | 0.413 | 5.857 | 0.354 |
| CoT-Evo | **0.729** | **8.230** | - | **0.704** | **7.847** | - |

pairwise comparisons between CoT-Evo and baseline CoTs, reporting win rates to capture relative quality. We do not report results for TwT, as it does not directly select CoTs with correct answers, leading to an unfair comparison. As shown in Table 2, CoT-Evo maintains over 70% usable data while producing CoTs that are more knowledge-rich and logically compact. Furthermore, even the strongest baseline achieves a win rate below 40%, demonstrating that its gains come not just from more data, but from better data.

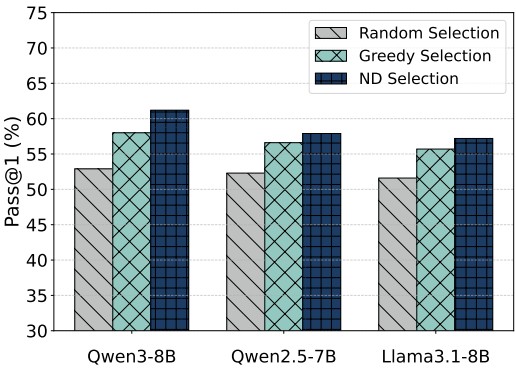 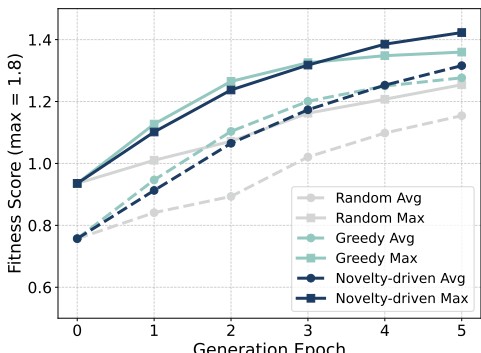

Figure 2: Comparison of selection strategies. **Left**: Impact of selection strategies on BioProBench (average Pass@1 across all tasks). **Right**: Fitness dynamics across generation epochs, showing both average and maximum fitness.

## 4.4 ABLATION STUDY

In this section, we explore the key components and hyperparameters of COT-EVO through a series of ablation studies. More analysis is provided in Appendix A2.2.

**Effectiveness of Selection Strategy.** To assess the effectiveness of our proposed novelty-driven selection (NDS) strategy, we compare it with two straightforward baselines: 1) *greedy selection*, which always chooses the candidates with the highest fitness score $\mathcal{R}$; and 2) *random selection*, which samples candidates uniformly at random. Average Pass@1 results on BioProBench are shown in Figure 2 (left). Across all three base models, NDS consistently surpasses the baselines. This trend can be explained by the evolution of fitness scores (Figure 2, right): random selection yields steady but slow improvements; greedy selection saturates within the first few epochs and often converges to local optima; in contrast, NDS drives faster and more stable gains, producing progressively higher-quality CoTs. Importantly, the budget of $B = 5$ remains well below the saturation point of COT-EVO, suggesting further room for improvement.

**Necessity of Recombination and Mutation.** To evaluate the effectiveness of the core modules for CoT optimization, we conduct ablation experiments by testing 1) COT-EVO without recombination, and 2) COT-EVO without mutation on BioProBench. As shown in Table 3,

Table 3: Ablation on recombination and mutation.

| Method | Qwen3-8B | Qwen2.5-7B | Llama3.1-8B |
|---|---|---|---|
| COT-EVO | **0.612** | **0.579** | **0.572** |
| w/o Recombination | 0.591 | 0.564 | 0.553 |
| w/o Mutation | 0.568 | 0.548 | 0.534 |

removing either module leads to a noticeable performance drop. However, the roles of recombination and mutation differ in important ways. Recombination primarily enhances the diversity of reasoning trajectories and promotes better utilization of knowledge; thus, removing this module degrades data quality and in turn weakens the performance of the student model. Mutation, on the other hand, is designed to correct errors and reduce redundant reasoning within CoTs. Without this module, the convergence of COT-EVO is hindered, resulting in fewer usable correct CoTs and consequently a more significant decline in student models' performance.

**Algorithm Scalability** We aim to validate whether the scale of our algorithm is a key factor affecting distillation performance. Specifically, Figure 3 investigates two major hyperparameters controlling the scale of COT-EVO: the iteration budget $B$ and the population size $n_{\text{pop}}$. As shown in Figure 3 (left), increasing the iteration budget initially leads to clear performance gains ($B \leq 3$), but these improvements gradually level off in later stages (in con-

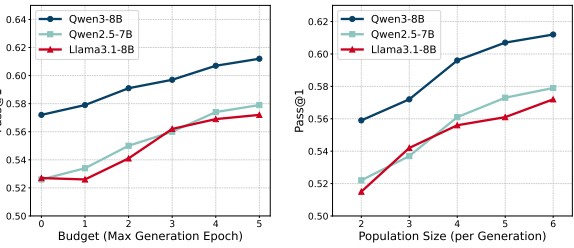

Figure 3: Impact of budget (left) and population size (right) on COT-EVO's distillation performance.

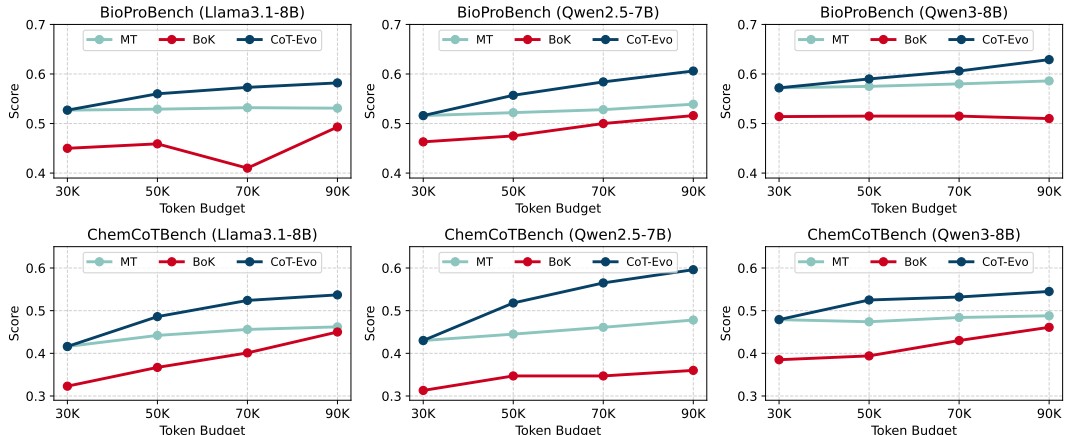

Figure 4: Performance comparison under equal token-budget constraints on BioProBench and ChemCoTBench across three LLMs, Llama3.1-8B-Instruct, Qwen2.5-7B-Instruct, and Qwen3-8B.

trast to Figure 2). This may be because we always select the highest-fitness CoTs from the candidate pool; after several epochs, correct and high-fitness CoTs are already obtained, reducing the marginal benefit of further iterations. A similar trend can be observed in the population-size curves (Figure 3, right). Empirically, when $n_{pop} < 5$, enlarging the population size enhances the diversity of the initial candidate pool at each iteration, which facilitates more effective recombination. Overall, increasing the scale consistently improves COT-EVO's performance, and near-optimal gains can be achieved under relatively modest budgets.

**Token-Budget Controlled Comparison.** To verify that the gains of COT-EVO do not simply stem from using more tokens, we further compare it with Multi-Teacher (MT) and Best-of-$K$ (BoK) under the same total distillation token budgets (30K, 50K, 70K, 90K). For BoK, we repeatedly sample CoTs from a single teacher until the budget is exhausted and then select the best CoT using the same fitness function as in Section 3.2, reporting the best result across all teachers. For MT, we cycle through all six teachers in rounds until the same budget is consumed, aggregating their CoTs and keeping the highest-fitness correct trajectory. For COT-EVO, we run the full evolutionary pipeline but terminate the loop immediately when the token budget is consumed, selecting the best candidate from the current population. As shown in Figure 4, COT-EVO consistently surpasses both MT and BoK at all budget levels, despite operating under identical token constraints. Notably, the performance gap is already clear in the low- and medium-budget regimes, indicating that evolutionary refinement can extract substantially more value per token than simple over-sampling and selection. As the budget grows, all methods improve, but COT-EVO maintains a stable advantage, suggesting that its gains come from systematically improving CoT quality rather than merely generating more trajectories. These results confirm that the evolutionary design of COT-EVO provides a more compute-efficient route to high-quality reasoning traces.

## 5 CONCLUSION

In this paper, we introduced COT-EVO, an evolutionary framework for chain-of-thought (CoT) distillation that goes beyond prior methods by performing intra-chain aggregation, integrating reasoning steps from multiple CoTs through an iterative evaluation–selection–variation–update process. This design yields distilled CoTs that are more accurate, rigorous, and diverse while remaining compact and efficient for training smaller scientific models. Extensive experiments show that COT-EVO consistently outperforms existing distillation approaches. Looking forward, we aim to evaluate its generalization across broader scientific domains, and to enhance the knowledge augmentation stage by incorporating external resources such as domain databases or structured knowledge graphs, making COT-EVO more scalable, knowledge-grounded, and practical for real-world scientific applications.

LIMITATIONS

While COT-EVO demonstrates strong performance in scientific reasoning distillation, it is not without limitations. First, leveraging multiple LLM thinkers introduces substantial computational and time costs, as generating and refining diverse trajectories requires repeated sampling and evaluation. This cost may hinder scalability to very large datasets or resource-constrained environments. Second, our evolutionary process currently relies on LLM-based reward signals to evaluate reasoning quality; while effective, these signals may still carry biases, and future work could explore integrating more verifiable reward signals (*e.g.*, rule-based) to further enhance robustness.

ACKNOWLEDGEMENTS

This work is funded by New Generation Artificial Intelligence - National Science and Technology Major Project (2025ZD0122803, K.D.), NSFC62301480 (K.D.), NSFC62302433 (Q.Z.), NS-FCU23A20496 (Q.Z.), and Ant Group Research Fund (K.D.).

ETHICS STATEMENT

This research aims to enhance the reliability of AI for scientific reasoning. Our experiments are conducted exclusively on public scientific benchmarks, which contain no sensitive or personally identifiable information. While we utilize pre-trained LLMs, our framework is designed to filter and refine their outputs to improve factual accuracy and logical rigor. We have made every effort to mitigate potential risks, such as the generation of incorrect scientific content, and therefore foresee no significant ethical issues arising from this work.

REPRODUCIBILITY STATEMENT

We are committed to the full reproducibility of our research. We will release our source code for the COT-EVO framework, fine-tuning scripts, and evaluation protocols. The paper provides comprehensive details on the public datasets, models, and hyperparameters used. Furthermore, all prompts required to replicate our evolutionary process are documented in the appendix. These resources will enable the community to verify our findings and build upon our work.

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

APPENDIX

## A1    ALGORITHM OF COT-EVO

**Algorithm 1:** COT-EVO: Evolutionary CoT Distillation Framework

**Input:** Training data $(x, y) \in \mathcal{D}_{\text{ori}}$, LLM thinkers $\mathcal{L}$, fitness function $\mathcal{R}$ ($s_{\text{EM}}$, $s_{\text{LEN}}$, and $s_{\text{KNOW}}$), budget $B$, population size $n_{\text{pop}}$
**Output:** Evolved CoT dataset $\mathcal{D}_{\text{evo}}$

```
// Initialization:
```
1   $\mathcal{P} \leftarrow \varnothing$;
2   **for** $(x, y) \in \mathcal{D}_{ori}$ **do**
3       obtain $\mathcal{K}_x$ via $\Theta$;
4       **for** $l_i \in \{l_1, \ldots, l_m\}$ **do**
5           $t_i \leftarrow l_i(x)$ (Eq. 1), $s_i \leftarrow \mathcal{R}(t_i)$ (Eq. 6), $\mathcal{P} \leftarrow \mathcal{P} \cup \{(t_i, s_i)\}$;
6       **end**
7       **for** $l_j \in \{l_{m+1}, \ldots, l_n\}$ **do**
8           $t_j \leftarrow l_j(x, \mathcal{K}_x)$ (Eq. 2), $s_j \leftarrow \mathcal{R}(t_j)$ (Eq. 6), $\mathcal{P} \leftarrow \mathcal{P} \cup \{(t_j, s_j)\}$;
9       **end**
10  **end**

11  **while** *budget $B > 0$ **and** $s \in \mathcal{P}$ not converged* **do**
12      $\mathcal{A} \leftarrow \text{NoveltySelect}(\mathcal{P})$ ;          // select parents (Sec. 3.3)
13      $\mathcal{O} \leftarrow \{\}$ ;          // offspring set
14      **for** $t_o \in \mathcal{A}$ **do**
15          draw operator op $\sim \text{Uniform}\{\text{Recombine}, \text{Mutate}\}$;
16          **if** op $= Recombine \wedge s_{EM}(t_o) = 0$ **then**
17              sample strategy provider $t_p \sim \mathcal{P}$;
18              $t_{\text{cand}} \leftarrow \mathcal{C}_r(t_o, t_p)$ ;        // recombination (Sec. 3.4)
19          **else**
20              choose mutation mode $m \sim \{\text{Add}, \text{Delete}, \text{Innovate}\}$;
21              $t_{\text{cand}} \leftarrow \mathcal{M}_u(t_o, m)$ ;        // Reflective mutation (Eqs. 14–16)
22          $s_{\text{cand}} \leftarrow \mathcal{R}(t_{\text{cand}})$ ;        // evaluate offspring
23          $\mathcal{O} \leftarrow \mathcal{O} \cup \{(t_{\text{cand}}, s_{\text{cand}})\}$;
24      **end**
25      $\mathcal{P} \leftarrow \mathcal{P} \cup \mathcal{O}$ ;        // Merge offspring into pool
26      keep top-$n_{\text{pop}}$ trajectories from $\mathcal{P}$ by fitness to maintain $|\mathcal{P}| \leq n_{\text{pop}}$;
27      $B \leftarrow B - 1$;
28  **end**

29  **return** $\mathcal{D}_{\text{evo}} = \{(x_i, t_i^\star = \arg\max_{t \in \mathcal{P}_{x_i}} \mathcal{R}(t), y_i)\}_{i=1}^N$.

## A2    ADDITIONAL RESULTS

### A2.1    RESULTS FOR INDIVIDUAL TEACHERS

In Section 4, we reported the performance of the *Single Teacher* (ST) baseline on BioProBench and ChemCoTBench (Table 1), as well as the reasoning quality of CoTs distilled under the ST setting (Table 2). These results were obtained by selecting the best-performing teacher as the representative. In this section, we provide the detailed results for all individual teachers. Specifically, Table A1 presents the performance of all teacher models across all tasks in both benchmarks, while Table A2 reports the quality evaluation of the training CoT data distilled from each teacher LLM.

As presented in Table A1, the two most powerful teacher models, Deepseek-R1 and Qwen3-235B-A22B-think, demonstrate a significant advantage. The student models distilled from these teachers outperform those from other teachers on the vast majority of scientific tasks. This suggests that larger and more capable LLMs, with their superior reasoning abilities, can generate more useful and higher-quality data during the distillation process. Consequently, the distilled CoTs exhibit more logical consistency and contain richer knowledge. Furthermore, the results reported in Table A2

Table A1: Performance comparison of CoT-Evo against baseline distillation methods (ST, MT, BoK) and original LLM teachers across BioProBench and ChemCoTBench. Best results are marked in **bold**, and second-best results are underlined.

| Teacher | BioProBench | | | | ChemCoTBench | | | | | |
|---|---|---|---|---|---|---|---|---|---|---|
| | PQA | ORD | ERR | | Und. | | | Edit | Reaction | Opt. |
| | Acc. ↑ | EM ↑ | Acc. ↑ | F1 ↑ | MAE↓ | TMS↑ | Acc.↑ | Acc.↑ | SR↑ | FTS↑ |
| *Base LLM: Llama3.1-8B-Instruct* | | | | | | | | | | |
| DeepSeek-R1 | 0.431 | 0.338 | 0.500 | 0.667 | 0.707 | 0.249 | 0.467 | 0.640 | 0.236 | 0.510 |
| Qwen3-32B-think | 0.424 | 0.308 | 0.586 | 0.641 | 0.955 | **0.260** | 0.487 | 0.635 | 0.269 | 0.469 |
| Qwen3-235B-A22B-think | 0.427 | 0.341 | 0.542 | 0.665 | 0.791 | 0.156 | 0.495 | 0.555 | 0.200 | 0.527 |
| Llama4-Scout-Backward | 0.429 | 0.193 | 0.505 | 0.553 | 1.276 | 0.151 | 0.482 | 0.512 | 0.182 | 0.506 |
| Llama4-Scout-CoT | 0.416 | 0.235 | 0.512 | 0.490 | 1.218 | 0.214 | 0.454 | 0.471 | 0.170 | 0.368 |
| Llama4-Scout-ToT | 0.431 | 0.208 | 0.558 | 0.521 | 1.095 | 0.173 | 0.460 | 0.539 | 0.196 | 0.435 |
| CoT-Evo | **0.512** | **0.419** | **0.657** | **0.698** | 0.375 | 0.250 | **0.639** | **0.707** | **0.340** | **0.597** |
| *Base LLM: Qwen2.5-7B-Instruct* | | | | | | | | | | |
| DeepSeek-R1 | 0.515 | 0.354 | 0.573 | 0.604 | 0.936 | 0.155 | 0.533 | 0.401 | 0.186 | 0.535 |
| Qwen3-32B-think | 0.506 | 0.331 | 0.593 | 0.584 | 0.982 | 0.146 | 0.500 | 0.426 | 0.177 | 0.414 |
| Qwen3-235B-A22B-think | 0.518 | 0.348 | 0.563 | 0.649 | 0.929 | 0.162 | 0.505 | 0.445 | 0.185 | 0.503 |
| Llama4-Scout-Backward | 0.516 | 0.289 | 0.571 | 0.536 | 1.104 | 0.129 | 0.495 | 0.358 | 0.174 | 0.538 |
| Llama4-Scout-CoT | 0.502 | 0.308 | 0.553 | 0.494 | 1.033 | 0.114 | 0.468 | 0.370 | 0.174 | 0.417 |
| Llama4-Scout-ToT | 0.506 | 0.274 | 0.580 | 0.568 | 1.069 | 0.138 | 0.490 | 0.374 | 0.182 | 0.465 |
| CoT-Evo | **0.551** | **0.448** | **0.675** | **0.671** | **0.497** | **0.306** | **0.664** | **0.602** | **0.332** | **0.577** |
| *Base LLM: Qwen3-8B-think* | | | | | | | | | | |
| DeepSeek-R1 | 0.601 | 0.368 | 0.569 | 0.532 | 0.726 | 0.204 | 0.558 | 0.606 | 0.327 | 0.554 |
| Qwen3-32B-think | 0.567 | 0.357 | 0.616 | 0.521 | 0.435 | 0.268 | 0.583 | 0.562 | 0.337 | 0.548 |
| Qwen3-235B-A22B-think | 0.547 | 0.361 | 0.525 | 0.534 | 0.405 | 0.183 | 0.520 | 0.550 | 0.375 | 0.599 |
| Llama4-Scout-Backward | 0.563 | 0.364 | 0.577 | 0.526 | 0.573 | 0.165 | 0.553 | 0.629 | 0.315 | 0.534 |
| Llama4-Scout-CoT | 0.546 | 0.327 | 0.562 | 0.475 | 0.674 | 0.214 | 0.568 | 0.602 | 0.302 | 0.525 |
| Llama4-Scout-ToT | 0.599 | 0.350 | 0.590 | 0.508 | 0.640 | 0.187 | 0.542 | 0.548 | 0.324 | 0.536 |
| CoT-Evo | **0.649** | **0.544** | **0.645** | **0.677** | **0.351** | **0.358** | **0.674** | 0.625 | **0.437** | **0.629** |

Table A2: Comparison of data utilization (Pass), reasoning quality (Quality) and win rate (WR) across distillation baselines and CoT-Evo.

| Teacher | BioProBench | | | ChemCoTDataset | | |
|---|---|---|---|---|---|---|
| | Pass | Quality | WR | Pass | Quality | WR |
| DeepSeek-R1 | 0.371 | 6.609 | 0.316 | 0.349 | 5.702 | 0.319 |
| Qwen3-32B-think | 0.261 | 6.184 | 0.286 | 0.275 | 4.699 | 0.290 |
| Qwen3-235B-A22B-think | 0.375 | 6.741 | 0.329 | 0.309 | 5.497 | 0.292 |
| Llama4-Scout-Backward | 0.192 | 5.610 | 0.218 | 0.178 | 5.225 | 0.257 |
| Llama4-Scout-CoT | 0.127 | 5.121 | 0.149 | 0.251 | 4.978 | 0.266 |
| Llama4-Scout-ToT | 0.166 | 5.119 | 0.165 | 0.217 | 4.516 | 0.218 |
| CoT-Evo | **0.729** | **8.230** | - | **0.704** | **7.847** | - |

confirm that these top-tier reasoning LLMs generally achieve higher data utilization (*i.e.*, pass rate) and superior CoT quality. In contrast, other LLM thinkers that rely on specific prompting strategies, such as Llama4-Scout-CoT, show a clear gap in data utilization, indicating that reasoning LLMs may be better at eliciting more comprehensive and complete reasoning.

An interesting nuance can be observed between the top two teachers. On BioProBench, DeepSeek-R1's reasoning quality and win rate are slightly lower than those of Qwen3-235B-A22B-think. We attribute this to DeepSeek-R1 occasionally generating redundant steps, as tasks related to experimental protocols prioritize the causal relationship between steps over the exploration of multiple possibilities. Conversely, for tasks requiring more fine-grained deliberation, such as molecule editing in ChemCoTBench, DeepSeek-R1 showcases richer ideas and more rigorous thinking, resulting in a higher overall quality score on that benchmark.

Table A3: Sensitivity of CoT-Evo to $k$ (k-nearest neighbors) on *Llama3.1-8B-Instruct*. Best results are marked in **bold**.

| Teacher | BioProBench | | | | ChemCoTBench | | | | | |
|---|---|---|---|---|---|---|---|---|---|---|
| | PQA | ORD | ERR | | | Und. | | Edit | Reaction | Opt. |
| | Acc. ↑ | EM ↑ | Acc. ↑ | F1 ↑ | MAE↓ | TMS↑ | Acc.↑ | Acc.↑ | SR↑ | FTS↑ |
| 1 | 0.508 | 0.395 | 0.638 | 0.681 | 0.427 | 0.230 | 0.631 | 0.653 | 0.339 | 0.567 |
| **2 (default)** | **0.512** | 0.419 | **0.657** | **0.698** | **0.375** | 0.250 | 0.639 | **0.707** | 0.340 | **0.597** |
| 3 | 0.502 | **0.426** | 0.648 | 0.693 | 0.405 | **0.348** | **0.669** | 0.684 | **0.348** | 0.595 |

Table A4: Sensitivity of CoT-Evo to $(\lambda_1, \lambda_2)$ on *Llama3.1-8B-Instruct*. Best results are marked in **bold**.

| Teacher | BioProBench | | | | ChemCoTBench | | | | | |
|---|---|---|---|---|---|---|---|---|---|---|
| | PQA | ORD | ERR | | | Und. | | Edit | Reaction | Opt. |
| | Acc. ↑ | EM ↑ | Acc. ↑ | F1 ↑ | MAE↓ | TMS↑ | Acc.↑ | Acc.↑ | SR↑ | FTS↑ |
| 0.1, 0.1 | 0.471 | 0.408 | 0.624 | 0.679 | 0.415 | **0.281** | **0.680** | 0.618 | 0.311 | 0.591 |
| **0.3, 0.1 (default)** | 0.512 | **0.419** | 0.657 | **0.698** | 0.375 | 0.250 | 0.639 | **0.707** | 0.340 | **0.597** |
| 0.5, 0.1 | 0.514 | 0.415 | 0.653 | 0.694 | 0.480 | 0.268 | 0.619 | 0.656 | **0.368** | 0.586 |
| 0.3, 0.2 | **0.525** | 0.410 | **0.681** | 0.668 | 0.417 | 0.257 | 0.607 | 0.680 | 0.349 | 0.567 |
| 0.3, 0.3 | 0.484 | 0.398 | 0.533 | 0.678 | **0.287** | 0.168 | 0.519 | 0.656 | 0.268 | 0.539 |

### A2.2 HYPERPARAMETER ABLATION STUDY

**Sensitivity to Population Size and Iteration Budget.** To assess the robustness of CoT-EVO to hyperparameter choices, we conduct an ablation study on the population size $n_{pop}$ and iteration budget $B$. Our default settings, such as $n_{pop} = 6$, are based on two principles: (i) matching the initial CoT sample size used in the multi-teacher (MT) baseline, so that the initial diversity is comparable without increasing cost in the first stage, and (ii) empirical evidence (Figure 3, left) suggesting that performance gains plateau when $n_{pop} \geq 5$ and $B \geq 3$. Beyond these points, performance improvement diminishes while computational cost rises sharply. This trade-off motivates our choice of $n_{pop} = 6$ as a practical balance between performance and cost. The results are shown in Table 3, where we observe that increasing $n_{pop}$ and $B$ improves performance but only yields marginal benefits after the saturation point.

**Sensitivity to $k$-Nearest Neighbors ($k$).** For the novelty-driven selection (NDS) strategy, we set the default $k = 2$ for the $k$-nearest neighbors parameter, as it provides a good balance between stability and discriminative power. Larger values of $k$ tend to overly smooth local structures, potentially overshadowing novel, low-frequency strategies, while smaller values (e.g., $k = 1$) make the distance metric overly sensitive to noise. As shown in Table A3, we experimented with $k \in \{1, 2, 3\}$ and observed minimal performance variation across different settings on both BioProBench and ChemCoTBench, supporting $k = 2$ as a robust and reproducible default choice.

**Sensitivity to Fitness Weights ($\lambda_1, \lambda_2$).** The fitness weights $\lambda_1$ and $\lambda_2$ in Eq. (6) reflect the relative importance of different evaluation criteria in our fitness function. As shown in Table A4, we performed a coarse grid search over $\lambda_1 \in \{0.1, 0.3, 0.5\}$ and $\lambda_2 \in \{0.1, 0.2, 0.3\}$, with results indicating that performance remains stable as long as $s_{EM}$ (exact-match correctness) is given the highest weight, and $s_{LEN}$ and $s_{KNOW}$ are given comparable smaller weights. When $\lambda_2$ is too large (e.g., 0.3), the model tends to select CoTs that use knowledge correctly but do not always reach the correct final answer, slightly hurting performance. Based on these findings, we chose $\lambda_1 = 0.3$ and $\lambda_2 = 0.1$ as a stable, low-sensitivity default.

### A2.3 GENERALIZATION TO NON-SCIENTIFIC DOMAINS

While our primary focus remains on enhancing LLM performance in specialized scientific reasoning tasks, we also acknowledge the potential for CoT-EVO to be applied in more general domains. To validate the generalizability of our approach, we conducted additional experiments on mathematical

Table A5: Comparison of model performance (pass1) across various general reasoning benchmarks. Best results are marked in **bold**.

| Task | base | ST | MT | BoK | LIMO | CoT-Evo |
|------|------|-----|-----|-----|------|---------|
| AIME24 | 13.33 | 23.08 | 20.00 | 25.00 | 10.00 | **45.00** |
| AMC23 | 47.50 | 70.59 | 57.89 | 75.00 | 75.00 | **85.71** |
| MATH500 | 75.40 | 80.11 | 86.67 | 89.20 | **91.80** | 88.80 |
| OlympiadBench | 36.89 | 52.17 | 64.66 | 67.87 | **70.75** | 66.14 |
| CHMath | 13.33 | 37.50 | 26.67 | 26.67 | 43.33 | **50.00** |
| Gaokao | 29.11 | 59.15 | 70.27 | 68.97 | 72.15 | **73.41** |
| Kaoyan | 30.15 | 41.14 | 44.81 | 40.74 | **47.24** | 46.10 |
| GradeSchool | 40.48 | 49.46 | 54.01 | **71.31** | 59.21 | 60.00 |
| Minerva | 49.63 | 48.36 | 49.46 | 51.03 | 58.22 | **60.54** |
| GPQA | 47.47 | 40.21 | 40.96 | 42.86 | 50.00 | **52.87** |
| AVG. | 38.33 | 50.18 | 51.73 | 59.64 | 62.19 | **64.21** |

and STEM tasks, using the LIMO dataset (Ye et al., 2025), a curated collection of 800 mathematical reasoning questions. Following the "Less is More" principle, LIMO's limited training data has demonstrated the ability to match the performance of OpenAI's o1-preview model (Jaech et al., 2024), which utilizes reinforcement learning.

In these experiments, we used the *Qwen2.5-7B-Instruct* model as the base model and compared several distillation baselines mentioned in Section 4.2.2: (1) the base model without training, (2) Single Teacher (ST), (3) Multi-Teacher (MT), and (4) Best-of-K (BoK). The results are summarized in Table A5. Notably, ST and MT approaches, due to lower data utilization rates (40.75% and 59.87%, respectively), showed significant performance gaps compared to CoT-Evo. On the other hand, BoK demonstrated a higher data utilization rate (78.5%) and notable improvements in performance, particularly in general reasoning tasks, highlighting the importance of more granular distillation strategies in non-scientific contexts.

Lastly, although the data utilization of LIMO and CoT-Evo was similar (100% vs. 95.75%), CoT-Evo outperformed LIMO in terms of reasoning quality. This underscores the crucial role of data quality in achieving better performance, with CoT-Evo demonstrating that the evolutionary refinement process can generate higher-quality CoTs, even when working with a dataset similar to that used in LIMO.

## A3    IMPLEMENTATION OF FINE-TUNING

In this work, to efficiently fine-tune LLMs, we leverage LLaMA-Factory (Zheng et al., 2024), Deep-Speed (Rasley et al., 2020) with ZeRO Stage 2 (Rajbhandari et al., 2020), and FlashAttention2 (Dao, 2023) across four NVIDIA A100 (80G) GPUs. We adopt AdamW (Loshchilov, 2017) as the optimizer with $\beta_1 = 0.9$, $\beta_2 = 0.95$, and a weight decay of 0.1. The peak learning rate is set to $2 \times 10^{-5}$ with 10% warm-up steps followed by cosine decay, a batch size of 32. We set the batch size to 32 and the maximum sequence length to 16,384. Training is conducted for 5 epochs to ensure optimal performance.

## A4    DETAILS OF COT QUALITY EVALUATION

While the student models distilled with CoT-Evo achieve the best overall performance, this does not *directly verify* the quality of the data generated by CoT-Evo. To comprehensively assess the quality of the synthesized CoTs, we adopt two complementary evaluation strategies: 1) **absolute scoring** of individual methods; and 2) **pairwise comparison** between CoT-Evo and baselines. These two strategies capture both absolute and relative perceptions of CoT quality. The compared distillation baselines include *Single Teacher* (ST), *Multi Teacher* (MT), and *Best-of-K* (BoK).

### A4.1 ABSOLUTE SCORING

Absolute scoring provides a quantitative measure of how CoTs perform across dimensions of interest. Motivated by the design and objectives of CoT-Evo, we define the following five dimensions for evaluating reasoning quality:

- **Knowledge Diversity**: Whether the model recalls or introduces sufficient concepts, theorems, or knowledge fragments to support reasoning or verification.

- **Knowledge Accuracy**: Whether the knowledge retrieved or cited in reasoning is factually correct, so as to avoid propagating errors into the student model.

- **Reasoning Diversity**: Whether the CoT explores meaningful alternative hypotheses or solution paths, promoting multi-perspective thinking and comprehensive understanding.

- **Logical Coherence**: Whether the CoT follows a clear, consistent, and error-free logical progression, avoiding excessive leaps or confusion.

- **Step Redundancy**: Whether unnecessary or repetitive steps are minimized, yielding concise yet complete reasoning.

For each CoT under evaluation, we employ GPT-5 as an LLM judge to assess performance along the above five dimensions and assign an overall integer score ranging from 0 to 10. The full prompt is provided in Prompt 1.

---

**Prompt 1: Prompt for Absolute Scoring**

You are a strict evaluator. Given a reasoning process that leads to the correct answer, your task is to evaluate the quality of this reasoning process across five dimensions: diversity of knowledge, accuracy of knowledge application, diversity of thinking, logical coherence, and redundancy of steps. Finally, provide a quality score.

[Reasoning Start]
{reasoning}
[Reasoning End]

You should strictly follow the instructions below:

1. First, identify all knowledge retrievals and citations within the reasoning chain. Then, leveraging your extensive knowledge and possibly external resources, judge their accuracy. Simultaneously, determine whether the knowledge involved in this reasoning chain is diverse enough to help students learn more. If there is no knowledge application in the reasoning chain, you can ignore this aspect during the evaluation.

2. Next, read the reasoning process step by step and consider whether the logic between each step is coherent (e.g., if there are causal relationships). At the same time, judge whether different thinking methods are employed in the reasoning process, which can contribute to students' thinking development.

3. Then, determine whether there are any unnecessary or redundant steps. These steps may merely repeat previous ones or be meaningless. If they exist, what is the proportion of redundant steps? You can make a rough estimate.

4. Finally, based on the analysis of the above five dimensions, provide an integer score between 0 and 10. A score of 0 means the reasoning chain has no learning value, indicating issues such as a large number of reasoning hallucinations, incorrect knowledge, and redundant steps. A score of 10 represents the highest quality reasoning chain, with accurate knowledge application, diverse thinking, coherent logic, and low step redundancy. As you are a strict evaluator, please give a score < 6 for reasoning processes with obvious flaws or those you're not satisfied with, and a score >= 6 for qualified reasoning processes.

Please carefully analyze the given reasoning process. First, output your evaluation process step by step, and then output an integer score between 0 and 10 on the last line. Please strictly follow the format below:
(Your evaluation process here)
[Result]Score[/Result]

---

A4.2   PAIRWISE COMPARISON

While absolute scores quantify quality in isolation, they may not fully capture relative improvements over baselines. To address this, we conduct pairwise comparisons between CoTs distilled by CoT-EVO and those generated by baseline methods.

For fairness, only CoTs with correct answers that were selected for training are included in the comparison. We again employ GPT-5 as the LLM judge. Specifically, GPT-5 is given a pair of CoTs (one from CoT-EVO, one from a baseline) and asked to decide which trajectory is superior overall based on the five evaluation dimensions. The prompt is detailed in Prompt 2. Importantly, GPT-5 is not allowed to declare a tie, *i.e.*, it must select the relatively better CoT. We then report the win rate of each baseline as the proportion of comparisons in which it was judged superior to CoT-EVO.

---

**Prompt 2: Prompt for Pairwise Comparison**

You are a strict evaluator. Given two reasoning processes, A and B, both of which lead to the correct answers, your task is to evaluate the quality of each reasoning process across five dimensions: knowledge diversity, accuracy of knowledge application, thinking diversity, logical coherence, and step redundancy. Finally, determine which reasoning process has higher quality (there will be no tie).

[Reasoning A Start]
{reasoning_a}
[Reasoning A End]

[Reasoning B Start]
{reasoning_b}
[Reasoning B End]

Please strictly follow the instructions below:

1. First, identify all knowledge retrievals and citations within each reasoning process. Then, leveraging your extensive knowledge and possibly external resources, determine which one has higher accuracy. Simultaneously, assess whether the knowledge involved in each reasoning chain is diverse enough to help students learn more.

2. Next, read the reasoning processes step by step and consider which one has more coherent logic between steps (e.g., causal relationships). Additionally, judge which reasoning process employs a wider variety of thinking methods, which can contribute to students' thinking development.

3. Then, identify if there are any unnecessary or redundant steps. These steps may merely repeat previous ones or be meaningless. If they exist, estimate which reasoning chain has fewer redundant steps.

4. Finally, based on the analysis of the above five dimensions, comprehensively determine which reasoning process, A or B, has higher quality. You should output either 0 or 1, where 0 indicates that A has higher quality and 1 indicates that B has higher quality.

Please carefully analyze the two given reasoning processes. First, output your evaluation process step by step, and then output an integer of 0 or 1 on the last line to represent your judgment. Please strictly follow the format below:
(Your evaluation process here)
[Result]0 or 1[/Result]

---

The results of these two evaluation strategies are reported in Table 2 of Section 4.3. Since CoT-EVO is not compared against itself, its win rate is omitted. CoT-EVO consistently achieves higher absolute scores, surpassing 8.0 on average, reflecting the robustness of its quality. Moreover, the baselines achieve win rates below 40% in pairwise comparisons, confirming the significant relative advantage of CoT-EVO.

## A5   PROMPTS FOR COT-EVO

This section details the prompts used in the evolutionary process of CoT-EVO, as described in Section 3. These prompts guide the LLMs to perform specific evaluation, recombination, or mutation tasks on the reasoning CoTs.

## A5.1 FITNESS FUNCTION PROMPTS

As described in Section 3.2, to evaluate the quality of candidate trajectories, we employ an LLM-as-a-Judge to assess the knowledge usage correctness.

**Prompt for knowledge usage correctness evaluation**  This prompt instructs the LLM judge to score a given CoT based on its application of the provided reference knowledge, returning a score from 1 to 5. The details of the prompt can be found in Prompt 3.

---

**Prompt 3: Prompt for Knowledge Usage Correctness Evaluation**

You are a meticulous evaluator specializing in scientific reasoning. Given a reasoning chain and a set of reference knowledge, your task is to evaluate the quality of the reasoning process based solely on how accurately and effectively it utilizes the provided knowledge. Finally, provide a quality score.

[Reasoning Chain Start]
{reasoning}
[Reasoning Chain End]

[Reference Knowledge Start]
{knowledge}
[Reference Knowledge End]

You should strictly follow the instructions below:

1. First, carefully read the provided reference knowledge to understand the key facts, principles, and data points. Then, read the entire reasoning chain to understand its overall logic and conclusion.
2. Next, compare the reasoning chain against the reference knowledge step-by-step. Assess the following: Is the knowledge used accurately and without misinterpretation? Are crucial pieces of knowledge from the reference correctly applied? Does the reasoning chain introduce any information or 'facts' that are not supported by the provided knowledge?
3. Finally, based on your analysis of the knowledge application, provide a single integer score between 1 and 5. A score of 1 indicates a complete disregard or fundamental misunderstanding of the reference knowledge. A score of 5 represents a perfect application of all relevant knowledge without any external hallucinations. You should adhere to the following scoring rubric:

- **Score 5 (Excellent):** The reasoning chain perfectly and accurately utilizes all relevant knowledge from the reference. No misinterpretations or unsupported claims are made.
- **Score 4 (Good):** The reasoning chain correctly uses most of the key knowledge, but may have very minor omissions or a slightly imprecise application that does not affect the overall logic.
- **Score 3 (Acceptable):** The reasoning chain uses the core knowledge from the reference, but with noticeable errors, omissions, or misinterpretations that weaken the argument.
- **Score 2 (Poor):** There is a significant misinterpretation or omission of key knowledge. The reasoning relies heavily on information outside the reference or gets crucial facts from the reference wrong.
- **Score 1 (Very Poor):** The reasoning chain completely ignores the reference knowledge, fabricates information, or demonstrates a fundamental misunderstanding of the provided facts.

Please carefully analyze the given reasoning process. First, output your evaluation process step by step, and then output an integer score between 1 and 5 on the last line. Please strictly follow the format below:
(Your evaluation process here)
[Result]Score[/Result]

---

## A5.2 COT RECOMBINATION PROMPTS

The goal of recombination is to create superior offspring trajectories that inherit and integrate the most effective strategies from different parent CoTs, promoting the fusion of diverse thoughts and knowledge.

**Prompt for Identifying the Binding Point**    This prompt tasks the recombiner model ($C_r$) with analyzing a strategy provider CoT ($t_p$) to identify the endpoint of the last reasonable thought, which serves as the crossover point. The specific prompt is detailed in Prompt 4.

---

**Prompt 4: Prompt for Identifying the Binding Point**

You are a rigorous scientific evaluator. When facing scientific problems, you excel at judging whether the chain of thought for solving the problem is sufficiently correct and logical, and can keenly identify the point where errors occur but are no longer corrected. Given a problem, its corresponding chain of thought (CoT), and the correct answer, your task is to preserve the longest reasonable prefix from the CoT (i.e., the correct or logically sound skeleton) and delete the portion starting from where errors or deviations from reasonable reasoning occur. Therefore, you need to locate the first sentence that exhibits logical errors or significant deviation from correct reasoning and output that sentence exactly as it appears (ensuring it completely matches the content in the given thought, without rewriting or summarizing).

[Query Start]
{query}
[Query End]

[CoT Start]
{thought}
[CoT End]

[Answer Start]
{answer}
[Answer End]

### Instructions

You must strictly adhere to the following instructions:

1. Carefully read and analyze the entire reasoning trajectory and logic of the thought, ultilizing the correct answer to judge the correctness and reasonableness of each step.

2. After analyzing all steps, carefully determine from which sentence the reasoning begins to exhibit obvious errors and the subsequent reasoning trajectory significantly deviates from the correct reasoning path (i.e., no longer returns to a reasonable reasoning trajectory). This sentence is the first sentence that needs to be deleted.

3. Note that if errors occur during reasoning but are eventually corrected through reflection or other means later, it indicates that the subsequent reasoning trajectory has not deviated, and **such errors can be ignored**!

4. You must ensure that the extracted deletion sentence exactly matches the original thought word-for-word, without any rewriting.

### Output Format

1. You should first analyze the thought step by step and output the judgment of step accuracy to identify the reasoning prefix to be preserved and the first sentence to be deleted. If the final answer of the given CoT does not match the correct answer, you must find the first sentence that deviates from the correct reasoning path, because there has at least one error.
2. At the end of your output, you should output the first sentence to be deleted in the following format for easy answer extraction:
[RESULT_START]
First sentence to be deleted
[RESULT_END]

---

**Prompt for Extracting Unique Information**    This prompt instructs the model to perform a comparative analysis between the strategy provider CoT ($t_p$) and the target CoT ($t_o$). The goal is to isolate the unique, valuable reasoning steps and knowledge that present only in the provider $t_p$. The specific prompt is detailed in Prompt 5.

---

**Prompt 5: Prompt for Extracting Unique Information**

You are an insightful scientific reviewer, adept at discovering and extracting key and useful **knowledge or information**. Given a scientific problem, a chain of thought representing the model's current progress (CoT_current), a chain of thought from external exploration (CoT_external), and the corresponding correct answer, your task is to organize and extract the guiding information or knowledge in CoT_external that can help optimize or improve the reasoning trajectory of CoT_current, based on the correct answer. This valuable information will be used to guide CoT_current to optimize and modify its own thinking process to facilitate obtaining the correct answer.

[Query Start]
{query}
[Query End]

[CoT Current Start]
{thought_current}
[CoT Current End]

[CoT External Start]
{thought_external}
[CoT External End]

[Correct Answer Start]
{answer}
[Correct Answer End]

### Instructions
You must strictly follow these instructions:

   1. The steps, information or final answer of CoT_external are partially incorrect. You should justify the correctness of them.

   2. Carefully read and analyze CoT_external, and identify **all** the totally correct knowledge or information that are missing or incorrectly analyzed in CoT_current based on the correct answer. You need to ensure this information is useful to guide CoT_current to optimize or improve its reasoning trajectory.

   3. Rewrite each knowledge or information extracted from CoT_external into a self-contained, generalizable sentence.

   4. Ensure that the extracted information is completely correct (please judge based on the correct answer), genuinely exists, and only exists in CoT_external. Ensure that there are no contradictions among the extracted information!!!

### Output Format

1. You should first highlight the correct answer, and then output the analysis of CoT_current and CoT_external to identify useful knowledge or information in CoT_external that is missing or incorrectly analyzed in CoT_current.
2. Next, you should judge the correctness of each knowledge or information extracted from CoT_external based on the correct answer. You should delete the incorrect knowledge or information after the judgement.
3. At the end of the output, you should **output the remaining extracted correct knowledge or information as a list** according to the following format:

[RESULT_START]
* knowledge/information-1
* knowledge/information-2
* ...
[RESULT_END]

---

**Prompt for Generating a New Trajectory**  This is the final synthesis step of recombination. The prompt guides the model to generate a new reasoning suffix, conditioned on the stable prefix from the target CoT $t_o$ and the unique information extracted from the provider $t_p$, thereby creating a novel, hybrid CoT. The specific prompt is detailed in Prompt 6.

---

**Prompt 6: Prompt for Generating a New Trajectory**

<think>
<tips>Before formally considering the user's question, let me reiterate my responsibilities: deeply analyze the user's inquiry, and set breakpoints when necessary to append additional correct information for subsequent reasoning. Breakpoints will be wrapped with "<breakpoint>" and "</breakpoint>", containing key information required for follow-up analysis. I will not treat the content within breakpoints as prior knowledge, but will revalidate or re-explain them during remaining reasoning steps. I will ensure contextual continuity before and after breakpoints. I will use detailed and rich thinking after each breakpoint to ensure the accuracy and comprehensiveness of my reasoning.</tips>

{prefix}

<breakpoint>
I have received the following correct information that I should use to continue my reasoning:

{breakpoint}

I should mention all the information in my follow-up reasoning. I should revalidate or re-explain the above correct information (within the "<breakpoint>") in my reasoning later. I should continue the reasoning from the previous context.
</breakpoint>

---

### A5.3 CoT Mutation Prompts

Mutation operators are designed for local optimization and error correction. They introduce targeted variations into a single CoT to fix flaws, enhance quality, and escape suboptimal reasoning patterns.

**Prompt for Additive Mutation** This prompt instructs the mutation model ($\mathcal{M}_u$) to enrich a given CoT by elaborating on its reasoning. It is used to add more logical details, deeper explanations, or relevant domain knowledge where the original CoT was underdeveloped. The specific prompt is detailed in Prompt 7.

---

**Prompt 7: Prompt for Additive Mutation**

You are a reasoning expert rich in professional knowledge and good at thinking. Given a query and a Chain of Thought (CoT), your task is to add more details to the CoT to make the thinking process more complete, coherent, and of higher quality. You should ensure that the final CoT includes more than {word} words.
[Query Start]
{query}
[Query End]

[CoT Start]
{thought}
[CoT End]

### Instructions

You must strictly adhere to the following instructions:
1. Please read the CoT sentence by sentence and identify assertions, assumptions, or conclusions that lack substantial evidence, collectively referred to as "unverified elements".
2. Leverage your extensive knowledge to add more evidence and details in the context of these unverified elements, using the same writing style as the original text. Make sure that the final CoT includes more than {word} words!
3. Ensure that no characters in the reasoning chain are modified, only adding evidence and details. 4. Output the optimized CoT from the beginning.
5. You should first output "[RESULT_START]", then directly output the optimized CoT, and finally

---

output "[RESULT_END]". Do not output other characters. The format is as follows:

[RESULT_START]
Optimized CoT, using the same format as the original CoT
[RESULT_END]

**Prompt for Deletive Mutation**  To improve the signal-to-noise ratio of the reasoning, this prompt instructs the model to prune a given CoT. It removes redundant steps, unproductive explorations, or extraneous information, resulting in a more concise and efficient trajectory for student model training. The specific prompt is detailed in Prompt 8.

---

**Prompt 8: Prompt for Deletive Mutation**

You are an insightful scientific critic, adept at identifying unnecessary steps and unusual words or sentences in a complete scientific reasoning process. Given a scientific query and its corresponding Chain of Thought (CoT), your task is to identify the core skeleton of the reasoning trajectory and remove abrupt words and sentences, and steps that are completely unnecessary, meaningless, and do not advance the reasoning further. Finally, directly output the remaining complete, high-quality, and clear reasoning trajectory. You should ensure that every character you output truly exists in the original text.

[Query Start]
{query}
[Query End]

[CoT Start]
{thought}
[CoT End]

### Instructions

You must strictly adhere to the following instructions:
1. First, carefully read the CoT to find abrupt words and sentences, and redundant and meaningless steps that do not advance the reasoning further. DO NOT delete valuable exploratory steps, or necessary information and knowledge obtained and retrieved in the middle.
2. It should be noted that since the CoT may reference correct answers, tips, or additional information provided by the user, although these external aids are necessary, the sources of this information, such as "the user says" or "the tips mention", should not appear in the CoT. Please remove these information sources (e.g., 'user', 'tips', 'correct answer') and treat the information as the model's internal knowledge. If removing these sources causes contextual incoherence, **you can add some details to ensure coherence**.
3. Then, directly output the remaining complete, high-quality, and contextually coherent reasoning trajectory from the beginning. You should ensure that every character you output truly exists in the original text, i.e., you cannot modify any characters of the given CoT except for the deleted sentences. Use the original format of the CoT.
4. A complete reasoning trajectory may at least includes 4 core elements: rephrase the query (DO NOT delete the first few sentences), find the solution step by step, validate the solution, and summarize the final result.
5. You should output "[RESULT_START]" at the beginning of your response, then directly output the remaining reasoning trajectory, and output "[RESULT_END]" at the end of your response. The format is as follows:

[RESULT_START]
Remaining reasoning trajectory here, using the same format as the original CoT
[RESULT_END]

---

**Prompt for Innovative Mutation**  This operator is a key mechanism for error correction in our framework. The prompt guides the model to first diagnose the logical fallacies in a faulty CoT by using the ground-truth answer as a reference, and to summarize them into corrective advice. We then

incorporate this advice into the original query to generate a new, correct reasoning trajectory that avoids the identified mistakes. The specific prompt is detailed in Prompt 9.

---

**Prompt 9: Prompt for Innovative Mutation**

Given a query, a relevant chain of thought (CoT), and the correct answer, your task is to check the correctness of each step in the CoT based on the correct answer and find all the critical errors that occur in the CoT. Finally, write each error as a one-sentence advice to prevent the error from recurring.

[Query Start]
{query}
[Query End]

[CoT Start]
{thought_current}
[CoT End]

[Correct Answer Start]
{answer}
[Correct Answer End]

### Instructions

Please strictly follow the following instructions:
1. First, analyze the correctness of each step in the CoT based on the correct answer.
2. Then, identify all the critical errors that occur in the CoT. This includes using incorrect knowledge, making wrong inferences, or having faulty intuitions.
3. Summarize each error into a one-sentence advice to prevent the same errors from recurring in subsequent attempts. **DO NOT mention the correct answer in your summary.**
4. Output all the summarized errors in a list, strictly following the format below:

[RESULT_START]
* advice-1-summary
* advice-2-summary
* ...
[RESULT_END]
5. Please output "[RESULT_START]" first, then directly output all the summarized advices, and finally output "[RESULT_END]".

---

## A6    TRANSITION KEYWORDS

In COT-EVO framework, particularly within the CoT Recombination described in Section 3.4, we need a systematic way to parse a continuous reasoning trajectory into a sequence of discrete, coherent *thoughts*. To achieve this, we utilize a predefined set of transition keywords. These keywords act as linguistic markers that signal a shift, pause, or pivot in the reasoning process. By identifying these keywords, our framework can effectively segment a long CoT into meaningful, modular thoughts. This segmentation is a critical prerequisite for identifying a suitable "binding point", enabling the model to perform more structured and meaningful recombination operations between different reasoning trajectories. Below is the list of transition keywords employed in COT-EVO.

---

**Transition Keywords in CoT**

```
"Wait", "But wait", "Alternatively", "However", "Let me", "Maybe", "Another", "Let's
see", "Backtrack", "Going back", "Okay", "Alright", "Hmm", "Hmmm", "Not sure", "Let
me double-check", "I think", "Good", "Got it", "That's correct"
```

---

## A7  USE OF LLMS

We acknowledge the use of generative AI in this work. Specifically, we employed LLMs to assist with response quality evaluation in Section 4.3 and to provide editorial support during the preparation of the manuscript.

## A8  CASE STUDIES

To provide a more granular understanding of COT-EVO's capabilities, this section presents two detailed case studies from distinct scientific domains: a biophysical protocol sequencing task (Section A8.1) and a complex chemical reaction prediction (Section A8.1). In both examples, we observe that even powerful teacher LLMs fail due to subtle yet critical flaws in their domain-specific knowledge or logical application. These cases are chosen to concretely illustrate how COT-EVO moves beyond simple distillation. By injecting more accurate domain knowledge and refining the logical flow, COT-EVO demonstrates its ability to synthesize a superior and correct reasoning path where the original teachers could not.

### A8.1  CASE STUDY ON SEQUENCING A PLANAR LIPID BILAYER PROTOCOL

This case study involves sorting the procedural steps of a complex biophysical experiment (Planar Lipid bilayer), where the correct order depends on understanding the physical and temporal constraints of the technique. The teacher models, DeepSeek-R1 and Qwen3-32B-think, both fail due to a shared logical error. They incorrectly assume that all physical apparatus, such as the `Curved Agar Bridges`, must be prepared before forming the delicate lipid bilayer. This reflects a superficial reasoning process that overlooks the practical vulnerabilities of the experimental setup.

In contrast, COT-EVO demonstrates a superior reasoning capability by introducing accurate, domain-specific knowledge about the experiment's constraints. It correctly identifies that the fragile lipid bilayer must be formed before the agar bridges are put in place to prevent physical disruption. By integrating this key piece of knowledge, COT-EVO improves the overall reasoning logic, correctly sequencing the prerequisites (preparing lipids and protein) and the assembly steps. This allows it to construct a coherent and correct workflow that respects the real-world constraints of the technique, showcasing its ability to refine flawed initial hypotheses into a logically sound solution.

---

**Case 1: Query**

To understand the role of the mitochondrial protein MPV17, whose mutations are linked to mitochondrial DNA depletion syndrome (MDS), the following protocol is used. The goal is to precisely measure the channel activity of the Drosophila melanogaster version of this protein (dMpv17) in a controlled environment using the planar lipid bilayer (PLB) technique.

Please sort the following procedural steps for this experiment in the correct order.
The steps are:
['dMPV17 Protein Preparation', 'PLB Formation', 'Preparation of Curved Agar Bridges', 'Measurement of dMpv17 Activity in the Presence of Uridine', 'Lipid Solution Preparation', 'Channel Activity Recording']

- Give me the correct order of the steps as a list of their original indices (start from 0), no other words.

---

**Correct Answer:** [4, 0, 1, 2, 5, 3]

---

**Case 1: Summarized reasoning trajectories of LLM thinkers**

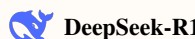 **DeepSeek-R1**

---

**Step 1: Initial Hypothesis on Core Setup**

My initial thought is to figure out the order of the core components. I know that to form the bilayer (1), I first need the lipid solution (4). Then I consider the curved agar bridges (2). *I reason that since the bridges are part of the physical setup, they should be prepared very early, perhaps even before the lipid solution is made.*

**Step 2: Incorporating Protein Preparation and First Re-evaluation**

Next, I try to fit the protein preparation (0) into the sequence. I get confused about its timing, initially thinking it might happen after the bilayer is already formed. This creates a logical conflict, as the protein must be ready *before* it can be added to the bilayer. I try to resolve this by considering a few different permutations of the setup steps (bridges, lipids, PLB) and the protein preparation.

**Step 3: Self-Correction on Protein and Experimental Steps**

I realize my previous thinking was flawed. It's clear that the protein must be prepared (0) before it is introduced into the system. It is also logical that you first have to record the baseline channel activity (5) before you can measure any changes caused by adding uridine (3), as you need a control to compare against.

**Step 4: Formulation of the Final (but flawed) Sequence**

With my corrected logic, I establish my final proposed order. *I conclude that the full process is: prepare the protein (0), prepare the agar bridges (2), prepare the lipid solution (4), and then form the PLB (1).* Following the setup, the experiment proceeds with recording baseline activity (5) and then measuring the effect of uridine (3). This leads me to the final, incorrect answer of [0, 2, 4, 1, 5, 3].

---

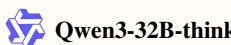 **Qwen3-32B-think**

**Step 1: Establishing the Initial Setup Order**

I started by recalling the general experimental flow. My first step was to order the setup components. *I decided that preparing the 'Curved Agar Bridges' (2) must be the very first step since they are part of the physical apparatus.* After that, I correctly reasoned that the 'Lipid Solution Preparation' (4) must come before 'PLB Formation' (1), because the solution is required to create the bilayer.

**Step 2: Sequencing Protein and Recording Steps**

Once the bilayer formation was placed, I moved on to the protein and measurement steps. *My key error was thinking that the 'dMPV17 Protein Preparation' (0) happens after the bilayer is already formed.* Following this step, I correctly ordered the data collection phase: first, the baseline 'Channel Activity Recording' (5), and then the specific experiment, 'Measurement of dMpv17 Activity in the Presence of Uridine' (3).

**Step 3: Final Consolidation and Review**

I concluded by putting all the steps together into a single sequence: bridges (2), lipids (4), PLB formation (1), protein prep (0), recording (5), and the uridine measurement (3). I did a final check, and this order seemed logical based on my flawed assumption about when the protein is prepared. This led me to finalize the incorrect answer [2, 4, 1, 0, 5, 3].

**Case1: Summarized reasoning trajectories of COT-EVO**

**Step 1: Establishing the Foundational Prerequisites**

I began by considering the standard workflow of a PLB experiment. I determined that the process must start with the preparation of the essential materials. *I identified 'Lipid Solution Preparation' (4) as the logical starting point because the lipid bilayer is the foundational structure of the entire experiment.*

Next, I placed 'dMPV17 Protein Preparation' (0) as a crucial, concurrent prerequisite, as the protein must be purified and correctly folded before any attempt to incorporate it into the membrane.

**Step 2: Constructing the Bilayer**

With the foundational materials ready, I reasoned that the next step is the formation of the planar lipid bilayer (PLB) itself (1). This critical process involves using the prepared lipid solution to create a stable, high-resistance membrane across the aperture in the experimental chamber, which will serve as the host for the protein channel.

**Step 3: Finalizing the Electrophysiological Setup**

After the delicate PLB is successfully formed, the next step is to finalize the setup for electrical recording. *Crucially, I placed 'Preparation of Curved Agar Bridges' (2) after the PLB is formed. This is because the agar bridges are part of the final recording circuit and adding them earlier could physically disrupt the fragile bilayer during its formation.* This corrects a major error present in the other reasoning traces.

**Step 4: Recording Baseline Channel Activity**

Once the bilayer is formed and the agar bridges are in place, the prepared protein can be introduced, allowing it to insert into the membrane. The subsequent step is to record the channel's intrinsic behavior. I identified this as 'Channel Activity Recording' (5), which serves as the essential baseline measurement against which any changes can be compared.

**Step 5: Performing the Experimental Measurement**

Finally, with the baseline activity established, the experiment concludes with the specific test condition. This involves adding uridine to the system and performing the 'Measurement of dMpv17 Activity in the Presence of Uridine' (3). I placed this step last because it is a specific experimental intervention designed to test the compound's effect on the channel's baseline function.

**Step 6: Consolidating the Final, Correct Sequence**

I summarized the entire logical flow: preparing the foundational materials (lipids, then protein), forming the bilayer, completing the electrical setup (agar bridges), recording a baseline, and finally testing the experimental variable. This step-by-step construction, respecting the physical and biological constraints of the technique, leads directly to the correct order: [4, 0, 1, 2, 5, 3].

A8.2  CASE STUDY ON CHEMICAL REACTION PREDICTION

In this case, we present a challenging reaction prediction task involving a **Sonogashira coupling**. Both powerful teacher LLMs, DeepSeek-R1 and Qwen3-235B-A22B-think, ultimately fail to predict the correct product. DeepSeek-R1 correctly identifies the reaction type but makes a critical error by misinterpreting the heteroaryl triflate as a diazonium salt, leading to an incorrect final structure. Qwen3-235B-A22B-think performs even more poorly, failing entirely to parse the key reactant and consequently giving up on the problem, returning a starting material as its answer.

In contrast, CoT-Evo demonstrates its ability to synthesize a superior reasoning path. It successfully inherits the crucial initial insight from the teachers—correctly identifying the reaction as a Sonogashira coupling. However, it then corrects their shared, fatal flaw by accurately identifying the triflate as the leaving group on the heteroaryl electrophile. By combining the correct foundational steps from its teachers with a new, pivotal insight, CoT-Evo not only avoids their errors but also creates a more accurate and enriched line of reasoning to arrive at the correct chemical product.

**Case 2: Query**

Given the reactants SMILES and reagents SMILES, your task is to predict the major product(s) SMILES using your experienced chemical Reaction Prediction knowledge.

Input:
Reactants SMILES separated by '.'
Reagents SMILES separated by '.'

Output: Major Product

The output SMILES must be valid and chemically reasonable. Do not provide any additional information beyond the requested SMILES strings. The answer should be a json format that includes the major product SMILES:

Question:
Reactants: C#Cc1ccccc1.Cl[Pd](Cl)([PH](c1ccccc1)(c1ccccc1)c1ccccc1)[PH](c1ccccc1)(c1ccccc1)c1ccccc1.O=c1ccc(OS(=O)(=O)C(F)(F)F)n[nH]1.[Cu]I
Reagents:

Answer:
{
"Major Product": ...
}

## Case 2: Summarized reasoning trajectories of LLM thinkers

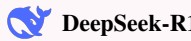 **DeepSeek-R1**

### Step 1: Initial Analysis and a Key Misidentification

I began by analyzing each reactant to understand its role. I correctly identified the phenylacetylene (`C#Cc1ccccc1`), the palladium complex, and the copper(I) iodide (`[Cu]I`). Based on these components, I correctly hypothesized that the reaction is likely a Sonogashira coupling. *However, I made a critical error in identifying the third reactant,* `O=c1ccc(OS(=O)(=O)C(F)(F)F)n[nH]1`. *I misinterpreted this molecule as a diazonium salt, focusing on the* `n[nH]` *moiety, instead of correctly identifying it as a heteroaryl triflate where the triflate group (`-OS(=O)(=O)C(F)(F)F`) is the actual leaving group.*

### Step 2: Exploring Consequences of the Flawed Premise

Based on my incorrect "diazonium salt" hypothesis, I explored various reaction mechanisms that were not relevant to the actual problem. I spent considerable effort trying to rationalize how a cyclic diazonium salt would decompose (e.g., by losing $N_2$ gas) and how the resulting species would couple with the alkyne. This led to significant confusion as I tried to parse the SMILES string and visualize a ring-opening or rearrangement that would not actually occur.

### Step 3: Acknowledging Contradictions and Reassessing the Goal

During my analysis, I noticed a contradiction in the prompt's instructions regarding whether to output the "Major Product" or "Byproduct". I correctly concluded that the primary goal was to predict the major product, as shown in the example answer format. I then returned to the chemical problem but was still hindered by my initial misidentification of the reactant, admitting that the structure was complex and confusing under my flawed assumption.

### Step 4: Formulating a Final Product Based on Misinterpretation

Despite the ongoing confusion, I proceeded to formulate a final answer. I maintained the general idea of a coupling reaction. *However, because I failed to recognize the triflate as the leaving group, my proposed product,* `O=c1ccc(OS(=O)(=O)C(F)(F)F)c(C#Cc2ccccc2)[nH]1`, *incorrectly kept the triflate group on the molecule. Furthermore, I incorrectly attached the phenylacetylene group to a carbon atom adjacent to the nitrogen in the ring, rather than at the position of the leaving group.*

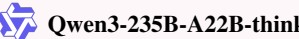 **Qwen3-235B-A22B-think**

**Step 1: Initial Analysis of Reactants**

I began by attempting to identify each component in the reaction. I correctly recognized phenylacetylene (`C#Cc1ccccc1`) and copper iodide (`[Cu]I`). I also identified the palladium species as a catalyst, though I expressed some minor confusion about the exact SMILES representation of its triphenylphosphine ligands.

**Step 2: Critical Failure to Interpret the Heteroaryl Electrophile**

My reasoning process completely stalled when I tried to parse the SMILES string `O=c1ccc(OS(=O)(=O)C(F)(F)F)n[nH]1`. *I became fixated on trying to name the specific heterocyclic ring system (speculating about triazoles, tetrazoles, etc.) and got lost trying to count the atoms to determine the ring size. Critically, I failed to identify the most important feature of this molecule for the reaction: the triflate group (`-OS(=O)(=O)C(F)(F)F`) as an excellent leaving group.* My inability to correctly interpret this key reactant prevented me from making any further progress on the reaction mechanism.

**Step 3: Abrupt and Unreasoned Conclusion**

My thought process was cut short while I was still struggling to understand the heteroaryl structure. Without providing any chemical justification or a logical path forward, I gave up on solving the problem. *My final answer,* `C#Cc1ccccc1`, *simply returns one of the starting materials, which incorrectly implies that no reaction takes place.*

---

### Case2: Summarized reasoning trajectories of COT-EVO

**Step 1: Initial Reactant Analysis and Reaction Hypothesis**

My first step is to break down the reactants to understand their roles. I identify `C#Cc1ccccc1` as phenylacetylene, the palladium complex as a catalyst, and `[Cu]I` as a co-catalyst. Based on this combination of a terminal alkyne, a palladium catalyst, and a copper co-catalyst, I correctly hypothesize that the reaction is a Sonogashira coupling.

**Step 2: Correctly Identifying the Electrophile and Leaving Group**

**Next, I analyze the key heteroaryl reactant, `O=c1ccc(OS(=O)(=O)C(F)(F)F)n[nH]1`. I correctly identify the crucial feature: the sulfonyloxy group (-OTf) is an excellent leaving group in cross-coupling reactions.** This singular, correct insight is the pivotal step that allows for a successful prediction.

**Step 3: Applying the Correct Mechanistic Principles**

I then consider the general mechanism for a Sonogashira coupling, which involves steps like oxidative addition of the palladium to the electrophile, transmetallation with the copper-activated alkyne, and reductive elimination. My reasoning correctly applies this mechanism to the actual electrophile, the heteroaryl triflate.

**Step 4: Deducing the Final Product Structure**

**From the correct application of the mechanism, I deduce that the reaction's outcome will be the replacement of the leaving group. The phenylacetylene group will form a new carbon-carbon bond at the exact position where the sulfonyloxy group was originally attached.**

**Step 5: Constructing and Confirming the Final Product SMILES**

Finally, I translate this structural understanding into the final SMILES string. By correctly replacing the `OS(=O)(=O)C(F)(F)F` group with the `C#Cc1ccccc1` group, I construct and confirm the valid, chemically

reasonable SMILES for the major product: `O=c1ccc(n[nH]1)C#Cc1ccccc1`. *This is equivalent to the correct answer*

