# OpenReview forum: "CoT-Evo: Evolutionary Distillation of Chain-of-Thought for Scientific Reasoning"
_ICLR.cc/2026/Conference — ICLR 2026 Poster_

### Official Review · Reviewer_7Nvu · 2025-10-26

**Soundness:** 2
**Presentation:** 3
**Contribution:** 2
**Rating:** 6
**Confidence:** 4

**Summary:**

This paper introduces CoT-Evo, an evolutionary CoT distillation framework for scientific reasoning. It generates diverse reasoning trajectories from multiple LLMs, enriches them with domain knowledge, and iteratively refines them using novelty-driven selection and recombination guided by a fitness function. The resulting dataset is used to fine-tune three compact-sized models, achieving strong performance on scientific reasoning benchmarks.

**Strengths:**

1. The paper is well-written and easy to follow.

2. It introduces an evolutionary framework where novelty-driven selection, reflective recombination, and mutation are used to iteratively refine reasoning trajectories, which is shown to be effective and well-motivated.

3. Fine-tuning three 7–8B models with the resulting dataset achieves strong performance, demonstrating the effectiveness of the proposed method.

**Weaknesses:**

1. Currently, the paper evaluates scientific reasoning only on BioProBench and ChemCoTBench. It would be helpful to include additional benchmarks, such as the well-known GPQA-diamond, to better demonstrate the effectiveness of the evolved CoT dataset.

2. The paper does not report the reasoning performance of the student model after each iteration of CoT evolution. Showing how the selected CoT data at each iteration contributes to model improvement would provide more insight into the co-evolution process.

3. CoT-Evo is specifically designed for scientific reasoning, but the paper does not evaluate its applicability to other complex reasoning domains, such as code reasoning, mathematical problem solving, or logical reasoning.

**Questions:**

See the weaknesses section

---

> ### Author Response · Authors · 2025-11-21
> **Responses to Reviewer 7Nvu (1)**
>
> Thank you for the valuable comment!
> > Currently, the paper evaluates scientific reasoning only on BioProBench and ChemCoTBench. It would be helpful to include additional benchmarks, such as the well-known GPQA-diamond, to better demonstrate the effectiveness of the evolved CoT dataset.
>
> We appreciate you highlighting GPQA-diamond. It is indeed one of the most established and influential scientific reasoning benchmarks. In this work, we focused primarily on BioProBench and ChemCoTBench because even state-of-the-art models still struggle on these tasks, and we were interested in examining whether CoT-Evo could deliver meaningful improvements where existing models perform poorly. Nevertheless, we fully agree that including GPQA-diamond would provide a more comprehensive assessment of the evolved CoT dataset.
>
> We will incorporate results on GPQA-diamond in the revision. If time permits, we will also share preliminary results during the discussion phase. Thank you again for the helpful suggestion.
>
> ---
>
> > The paper does not report the reasoning performance of the student model after each iteration of CoT evolution. Showing how the selected CoT data at each iteration contributes to model improvement would provide more insight into the co-evolution process.
>
> Thank you for raising this question. We agree that reporting the student model’s performance after each evolution iteration would provide more insight into the co-evolution dynamics. While running a full fine-tuning cycle at every iteration is computationally prohibitive, **we provide two forms of indirect but informative evidence that approximate the effect of early iterations.**
>
> First, Table R1 presents results under different token budgets for CoT-Evo and baseline methods. Although token consumption per iteration is not fixed and cannot be precisely aligned to iteration boundaries, these budget-controlled settings naturally form a set of coarse-grained snapshots of the evolutionary process. Smaller token budgets correspond to early stopping points where only the initial or early-evolution CoTs have been generated; larger budgets correspond to later stages where more refined candidates have emerged. Under this interpretation, we observe that even at low budgets where evolution has only partially unfolded, CoT-Evo already surpasses Best-of-K and Multi-Teacher, **indicating that early iterations produce meaningfully stronger CoTs than those available at initialization.**
>
> Second, the fitness-trajectory analysis (Figure 2 right) shows that the average and best fitness scores within the candidate pool steadily improve over iterations. Since higher-fitness CoTs correlate with stronger downstream student performance (Table 2), this monotonic improvement strongly suggests that each iteration contributes increasingly higher-quality reasoning traces, even without explicitly training a student model at each intermediate step.
>
> ---
>
> **Table R1. Performance under equal token-budget constraints across different models.**
> | Model               | 30K tokens             | 50K tokens            | 70K tokens           | 90K tokens            |
> | ------------------- | --------------- | --------------- | --------------- | --------------- |
> | llama3.1-8b-MT      | 0.527/0.416     | 0.529/0.442     | 0.532/0.456     | 0.531/0.462     |
> | llama3.1-8b-BoK     | 0.450/0.323     | 0.459/0.367     | 0.410/0.401     | 0.493/0.450     |
> | llama3.1-8b-CoT-Evo | 0.527/0.416 | **0.560/0.486** | **0.573/0.524** | **0.582/0.537** |
> | qwen2.5-7b-MT       | 0.516/0.430     | 0.522/0.445     | 0.528/0.461     | 0.539/0.478     |
> | qwen2.5-7b-BoK      | 0.463/0.313     | 0.475/0.347     | 0.500/0.347     | 0.516/0.360     |
> | qwen2.5-7b-CoT-Evo  | 0.516/0.430 | **0.557/0.518** | **0.584/0.565** | **0.606/0.596** |
> | qwen3-8b-MT         | 0.572/0.479     | 0.575/0.474     | 0.580/0.484     | 0.586/0.488     |
> | qwen3-8b-BoK        | 0.514/0.385     | 0.515/0.394     | 0.515/0.430     | 0.510/0.461     |
> | qwen3-8b-CoT-Evo    | 0.572/0.479 | **0.590/0.525** | **0.606/0.532** | **0.629/0.545** |

---

> ### Author Response · Authors · 2025-11-21
> **Responses to Reviewer 7Nvu (2)**
>
> Thank you for the thoughtful comment.
> > CoT-Evo is specifically designed for scientific reasoning, but the paper does not evaluate its applicability to other complex reasoning domains, such as code reasoning, mathematical problem solving, or logical reasoning.
>
> CoT-Evo is indeed designed with scientific reasoning as the primary target domain. Our main objective in this paper is to address the unique challenges of scientific reasoning, such as scarce high-quality long-form CoTs, complex domain-specific knowledge requirements, and limited verifiable intermediate signals, which are less pronounced in other reasoning domains like math, logic, or code. For these reasons, our evaluations prioritize scientific tasks where the need for improved reasoning quality is most acute.
> That said, the evolutionary framework itself is general and does not rely on science-specific assumptions. We fully agree that understanding its broader applicability is valuable. We are currently running CoT-Evo on additional non-scientific reasoning benchmarks (e.g., mathematical or logical reasoning), and we will provide the corresponding results within the next few days to demonstrate its generalization beyond scientific tasks.

---

### Official Review · Reviewer_pgWS · 2025-10-30

**Soundness:** 3
**Presentation:** 2
**Contribution:** 3
**Rating:** 6
**Confidence:** 3

**Summary:**

This paper addresses the problem in COT distillation for scientific reasoning: advanced LLM teachers often produce flawed reasoning. Distilling from these flaws limits student model performance.

To solve this, the authors propose COT-Evo, a novel "Evolutionary CoT Distillation" framework. The core idea is to actively "evolve" and "synthesize" higher-quality reasoning paths rather than just selecting existing ones.

The framework initializes a diverse pool of CoTs from multiple "LLM Thinkers." It then iteratively optimizes this pool using an evolutionary loop: 1) Fitness Evaluation (correctness, length, knowledge use), 2) Novelty-Driven Selection (balancing quality and diversity), and 3) Recombination/Mutation (integrating strengths and fixing errors). This process generates a high-fidelity "evolved" dataset used to fine-tune student models.

Experimental results show that models trained on the COT-Evo dataset achieve better performance on challenging scientific benchmarks (BioProBench and ChemCoTBench).

**Strengths:**

Creatively applies evolutionary algorithms and novelty-driven selection to synthesize superior reasoning, moving beyond the traditional "select-the-best" paradigm. Clearly identifies and offers a promising solution to the critical problem of distilling from "fallible" teachers in complex scientific domains.

Robust validation on challenging benchmarks with comprehensive baselines. In-depth analysis and ablations effectively prove that performance gains come from higher-quality data, not just increased sampling.

**Weaknesses:**

The framework is computationally intensive, requiring multiple LLM thinkers and repeated LLM calls within the evolutionary loop. The practical overhead compared to baselines is not fully quantified.

The framework relies on an LLM for critical steps (fitness, recombination, mutation, and quality evaluation), creating a potential point of failure if the judge is biased or flawed.

The evolutionary process may be sensitive to hyperparameters (e.g., a small population size $n_{pop}=6$ and $k=2$). The justification for these specific choices is limited.

**Questions:**

Q1: The COT-Evo framework appears significantly more computationally intensive (e.g., in total tokens or LLM calls) than baselines like MT or BoK, given the iterative evaluation and generation loop. It would greatly strengthen the paper to include a quantitative discussion on this. For instance, could you provide a comparison of the approximate token consumption required to generate the dataset versus the baselines? Alternatively, an analysis of performance under an equivalent, fixed token budget would be very insightful for assessing the method's practical efficiency.

Q2: While Figure 3 helpfully explores the budget ($B$) and population size ($n_{pop}$), could you provide more justification for the selection of other key parameters, such as the fitness weights ($\lambda_1, \lambda_2$) and the $k$-nearest neighbors value ($k=2$)? Even a brief explanation of how these were chosen, or a discussion of their sensitivity, would be valuable for reproducibility.

Q3: Since the "Recombination" operation is well-defined as a "guided generation" process (using a prefix from $t_o$ and information from $t_p$), the analogy to a traditional EA "crossover" might not be necessary and could potentially cause confusion for readers familiar with classic EA terminology. Clarifying it simply as a novel, guided operator might be more straightforward.

---

> ### Author Response · Authors · 2025-11-21
> **Responses to Reviewer pgWS (1)**
>
> Thank you for your insightful comments!
> > The framework is computationally intensive, requiring multiple LLM thinkers and repeated LLM calls within the evolutionary loop. The practical overhead compared to baselines is not fully quantified.
>
> > Q1: The COT-Evo framework appears significantly more computationally intensive (e.g., in total tokens or LLM calls) than baselines like MT or BoK, given the iterative evaluation and generation loop. It would greatly strengthen the paper to include a quantitative discussion on this. For instance, could you provide a comparison of the approximate token consumption required to generate the dataset versus the baselines? Alternatively, an analysis of performance under an equivalent, fixed token budget would be very insightful for assessing the method's practical efficiency.
>
> Thank you for bringing up this important concern regarding computational efficiency. We fully acknowledge that CoT-Evo introduces additional LLM calls due to its iterative evaluation–selection–variation loop. To rigorously quantify this overhead, **we conducted an additional analysis based on token-budget–controlled experiments, summarized in Table R1.** This setting enforces exactly the same total number of tokens across CoT-Evo and all baselines, thereby isolating the effect of evolutionary refinement from pure compute scaling.
>
> Under this equal-compute regime, standard baselines such as Multi-Teacher (MT) and Best-of-K (BoK) consume the entire token budget solely through forward sampling: MT repeatedly cycles through its teachers until the budget is used up, while BoK samples K trajectories from a single teacher and selects the best one. In contrast, CoT-Evo allocates part of the same budget to evolutionary refinement (recombination, mutation, and re-evaluation), and immediately terminates the loop once the shared token limit is reached. Thus, all methods consume identical levels of compute, but differ fundamentally in how the tokens are used.
> The results show two key findings.
>
> First, **CoT-Evo achieves substantially higher performance than MT and BoK under identical token budgets.** Even though the baselines can sample many more raw trajectories, their gains remain small because additional samples cannot repair or restructure flawed reasoning. CoT-Evo, however, uses part of the budget to refine reasoning at the structural level—identifying binding points, integrating correct steps from alternative trajectories, and performing reflective revisions—which yields disproportionately higher-quality CoTs.
>
> Second, **token-efficiency trends in Table R1 reveal that CoT-Evo is most compute-efficient in the low- and mid-budget regimes (≤50k tokens)**, where performance grows rapidly; beyond this, the gains taper but remain positive, indicating that CoT-Evo has not saturated and continues to benefit from additional reasoning refinement.
>
> In summary, although CoT-Evo introduces additional LLM calls, the token-controlled experiments demonstrate that its improvements cannot be attributed to consuming more compute. Instead, the evolutionary loop provides a qualitatively different advantage—systematically improving the internal reasoning structure rather than merely sampling more of it. We will include these quantitative comparisons and clarifications in the revised version to make the computational trade-offs fully transparent.
>
> ---
>
> **Table R1. Performance under equal token-budget constraints across different models.**
> | Model               | 30K tokens             | 50K tokens            | 70K tokens           | 90K tokens            |
> | ------------------- | --------------- | --------------- | --------------- | --------------- |
> | llama3.1-8b-MT      | 0.527/0.416     | 0.529/0.442     | 0.532/0.456     | 0.531/0.462     |
> | llama3.1-8b-BoK     | 0.450/0.323     | 0.459/0.367     | 0.410/0.401     | 0.493/0.450     |
> | llama3.1-8b-CoT-Evo | 0.527/0.416 | **0.560/0.486** | **0.573/0.524** | **0.582/0.537** |
> | qwen2.5-7b-MT       | 0.516/0.430     | 0.522/0.445     | 0.528/0.461     | 0.539/0.478     |
> | qwen2.5-7b-BoK      | 0.463/0.313     | 0.475/0.347     | 0.500/0.347     | 0.516/0.360     |
> | qwen2.5-7b-CoT-Evo  | 0.516/0.430 | **0.557/0.518** | **0.584/0.565** | **0.606/0.596** |
> | qwen3-8b-MT         | 0.572/0.479     | 0.575/0.474     | 0.580/0.484     | 0.586/0.488     |
> | qwen3-8b-BoK        | 0.514/0.385     | 0.515/0.394     | 0.515/0.430     | 0.510/0.461     |
> | qwen3-8b-CoT-Evo    | 0.572/0.479 | **0.590/0.525** | **0.606/0.532** | **0.629/0.545** |

---

> ### Author Response · Authors · 2025-11-21
> **Responses to Reviewer pgWS (2)**
>
> Thank you for your valuable comment!
> > The framework relies on an LLM for critical steps (fitness, recombination, mutation, and quality evaluation), creating a potential point of failure if the judge is biased or flawed.
>
> We appreciate the reviewer highlighting the potential risk of relying on an LLM judge. We would like to clarify that **CoT-Evo does not depend on a single black-box judge to make all critical decisions.** Instead, the LLM-based evaluation is embedded within a multi-signal constrained framework. Our fitness function comprises three components: (i) exact-match correctness, computed by scripts or task-specific evaluators; (ii) a length appropriateness prior derived from scientific reasoning corpora; and (iii) knowledge-usage correctness, which is the only part using an LLM-as-a-judge. Even here, the judge evaluates a trajectory relative to an explicit knowledge snippet $K_x$ —not through open-ended subjective scoring. Thus, the judge acts more like a knowledge-alignment checker with a constrained evaluation scope, rather than the sole decision-maker of the evolutionary loop.
>
> Recombination and mutation also involve LLM calls, but their inputs are highly structured:
> * a retained prefix that has already passed fitness filtering,
> * key steps or knowledge fragments extracted from other trajectories, and
> * explicit editing instructions (e.g., “add logical details,” “fix incorrect reasoning”).
>
> Thus, **the LLM is not generating free-form reasoning from scratch; it performs localized refinement over grounded scientific content, and every generated variant is again filtered by the next fitness evaluation (including rule-based exact-match checks).**
>
> Finally, we fully acknowledge that judge bias is a potential failure mode. In the revised manuscript, we will explicitly discuss this limitation and include two mitigation directions: (i) multi-judge or cross-judge evaluation, reducing reliance on any single model; and (ii) additional robustness results showing the stability of CoT-Evo under different judge models or weighting schemes. Overall, the current use of an LLM judge is limited, controlled, and complemented by strong script-based signals and explicit knowledge constraints, ensuring that CoT-Evo benefits from LLM evaluation without becoming overly dependent on a potentially biased judge.
>
> ---
>
> Thank you for your comment!
> > Q3: Since the "Recombination" operation is well-defined as a "guided generation" process (using a prefix from t_o and information from t_p), the analogy to a traditional EA "crossover" might not be necessary and could potentially cause confusion for readers familiar with classic EA terminology. Clarifying it simply as a novel, guided operator might be more straightforward.
>
> We appreciate you highlighting this terminology concern. We agree that framing our “Recombination” operator strictly as a classical evolutionary crossover may introduce unnecessary confusion, since our mechanism is more accurately described as a *guided generation process* that uses a retained prefix from t_o and selected information from t_p. We are happy to clarify this in the revision and will rephrase it as a novel, guided operator rather than relying on traditional EA terminology. Thank you again for the helpful suggestion.

---

> ### Author Response · Authors · 2025-11-21
> **Responses to Reviewer pgWS (3)**
>
> Thank you for your comments!
> > The evolutionary process may be sensitive to hyperparameters (e.g., a small population size n_pop=6 and k=2). The justification for these specific choices is limited.
>
> > Q2: While Figure 3 helpfully explores the budget (B) and population size (n_pop), could you provide more justification for the selection of other key parameters, such as the fitness weights (λ1,λ2) and the k-nearest neighbors value (k)? Even a brief explanation of how these were chosen, or a discussion of their sensitivity, would be valuable for reproducibility.
>
> Our goal in COT-Evo is to build an evolutionary framework that (i) runs with low additional overhead and (ii) is robust to reasonable hyperparameter choices. For the population size $n_{\text{pop}}$ and the number of iterations (B), our default settings (e.g., $n_{\text{pop}} = 6$) follow two principles: first, they match the initial CoT sample count used in the multi-teacher (MT) baseline, so the initial diversity is comparable without increasing cost in the first stage; second, as shown in Figure 3, performance gains on scientific reasoning tasks start to saturate when $n_{\text{pop}} \ge 5$ and $B \ge 3$. Beyond this point, increasing population size or iterations yields diminishing returns while compute grows rapidly. **Hence we choose $n_{\text{pop}} = 6$ as a practical trade-off between performance and cost.**
>
> For the k-nearest neighbors parameter $k$ in novelty search, we use a small local neighborhood (default $k = 2$). Conceptually, Novelty Search with Local Competition relies on local behavioral differences to measure diversity. **A large $k$ tends to overly smooth local structure, causing genuinely novel but low-frequency strategies to be overshadowed, while too small a $k$ (e.g., $k=1$) makes the distance metric overly sensitive to noise from a single neighbor.** Given that our behavior embeddings are high-dimensional LLM representations, we found $k = 2$ to offer a good balance between stability and discriminativeness in practice. In preliminary experiments, we tested $k \in \{ 1, 2, 3 \} $ and **observed only minor performance variation on both scientific benchmarks (see Table R4)**, so we adopt $k = 2$ as a simple and robust default. We will add this qualitative explanation and the table to the revised version for better reproducibility.
>
> Regarding the fitness weights $\lambda_1, \lambda_2$, their design reflects the structure and relative reliability of the signals. The term $s_{\text{EM}})$ (exact-match correctness) is the most reliable, executable metric and thus implicitly has the highest influence; $s_{\text{LEN}}$ encodes a structural prior on “reasonable information density” in scientific CoTs; $s_{\text{KNOW}}$ checks knowledge alignment relative to an explicit snippet $K_x$. **We performed a coarse grid search over $\lambda_1 \in \{0.1,0.3,0.5\}, \lambda_2 \in \{0.1, 0.2, 0.3\}$ on llama3.1-8b-instruct (Table R2).** As long as the ordering “$s_{\text{EM}}$ dominant, $s_{\text{LEN}}$ and $s_{\text{KNOW}}$ with comparable smaller weights” is respected, performance changes are small. When $\lambda_2$ is too large (e.g., 0.3), we observe that the model starts to over-select CoTs that *use* knowledge correctly but do not always reach the correct final answer, slightly hurting task performance. Based on this trade-off, we choose $\lambda_1 = 0.3, \lambda_2 = 0.1)$ as a stable, low-sensitivity default that is easy to reproduce.
>
> In the revision, we will add these explanations and sensitivity summaries to further improve clarity and reproducibility.

---

> ### Author Response · Authors · 2025-11-21
> **Responses to Reviewer pgWS (4)**
>
> ---
>
> **Table R2. Sensitivity of CoT-Evo to $(\lambda_1, \lambda_2)$ on llama3.1-8b-instruct**
>
> | $(\lambda_1, \lambda_2)$ | **BioProBench PQA** Acc ↑ | **BioProBench ORD** EM ↑ | **BioProBench ERR** Acc ↑ | **BioProBench ERR** F1 ↑ | **ChemCoT Und.** MAE ↓ | **ChemCoT Und.** TMS ↑ | **ChemCoT Und.** Acc ↑ | **ChemCoT Edit** Acc ↑ | **ChemCoT Reaction** SR ↑ | **ChemCoT Opt.** FTS ↑ |
> | --| -- | -- | -- | -- | -- | -- | -- | -- | -- | -- |
> | 0.1, 0.1 | 0.471 | 0.408 | 0.624 | 0.679 | 0.415 | **0.281** | **0.680** | 0.618 | 0.311 | 0.591 |
> | **0.3, 0.1 (default)** | 0.512 | **0.419** | 0.657 | **0.698** | 0.375 | 0.250 | 0.639 | **0.707** | 0.340 | **0.597** |
> | 0.5, 0.1 | 0.514 | 0.415 | 0.653 | 0.694 | 0.480 | 0.268 | 0.619 | 0.656 | **0.368** | 0.586 |
> | 0.3, 0.2 | **0.525** | 0.410 | **0.681** | 0.668 | 0.417 | 0.257 | 0.607 | 0.680 | 0.349 | 0.567 |
> | 0.3, 0.3 | 0.484 | 0.398 | 0.533 | 0.678 | **0.287** | 0.168 | 0.519 | 0.656 | 0.268 | 0.539 |
> ---
> **Table R4. Sensitivity of CoT-Evo to $k$ (k-nearest neighbors) on llama3.1-8b-instruct**
>
> | (k) | **BioProBench PQA** Acc ↑ | **BioProBench ORD** EM ↑ | **BioProBench ERR** Acc ↑ | **BioProBench ERR** F1 ↑ | **ChemCoT Und.** MAE ↓ | **ChemCoT Und.** TMS ↑ | **ChemCoT Und.** Acc ↑ | **ChemCoT Edit** Acc ↑ | **ChemCoT Reaction** SR ↑ | **ChemCoT Opt.** FTS ↑ |
> | -- | -- | ------------------------ | ------------------------- | ------------------------ | ---------------------- | ---------------------- | ---------------------- | ---------------------- | ------------------------- | ---------------------- |
> | 1 | 0.508 | 0.395 | 0.638 | 0.681 | 0.427 | 0.230 | 0.631 | 0.653 | 0.339 | 0.567 |
> | **2 (default)** | **0.512** | 0.419 | **0.657** | **0.698** | **0.375** | 0.250 | 0.639 | **0.707** | 0.340 | **0.597** |
> | 3 | 0.502 | **0.426** | 0.648 | 0.693 | 0.405 | **0.348** | **0.669** | 0.684 | **0.348** | 0.595 |

---

> ### Comment · Reviewer_pgWS · 2025-11-28
>
> I appreciate the authors' detailed response. Since the rebuttal has effectively addressed my questions, I will keep my original rating.

---

### Official Review · Reviewer_tiKN · 2025-11-01

**Soundness:** 2
**Presentation:** 3
**Contribution:** 2
**Rating:** 6
**Confidence:** 3

**Summary:**

Existing CoT distillation methods in scientific domains are vulnerable to issues of faulty reasoning and knowledge deficiency. Prior approaches either perform single-chain compression/pruning or select a single chain from multiple teachers, lacking the ability to refine and integrate reasoning at a fine-grained level. This paper proposes an evolutionary CoT distillation framework, COT-EVO, which emphasizes intra-chain recombination and mutation rather than merely inter-chain selection.

**Strengths:**

Proposes a novel evolutionary CoT distillation framework that introduces the concept of genetic algorithms into reasoning-path optimization. Demonstrates improved performance of small models in scientific reasoning tasks, with ablation studies verifying the effectiveness of the recombination, mutation, and novelty selection components.

**Weaknesses:**

The authors did not conduct direct comparisons with recent process-level distillation or evolutionary reasoning methods such as TwT, Retro-Search, and BREAD, leaving the unique contribution of COT-EVO unclear.

The paper does not compare its method with standard CoT under equal compute (equal token count) conditions.

Some of the reported performance improvements are not statistically significant.

The method is only validated on small-scale models, and it remains uncertain whether the gains would persist for larger models.

**Questions:**

See weakness.

---

> ### Author Response · Authors · 2025-11-21
> **Responses to Reviewer tiKN (1)**
>
> Thank you for the valuable comment!
> > The authors did not conduct direct comparisons with recent process-level distillation or evolutionary reasoning methods such as TwT, Retro-Search, and BREAD, leaving the unique contribution of COT-EVO unclear.
>
> We appreciate your suggestion to compare against more recent process-level distillation and evolutionary reasoning approaches. In response, **we include two additional baselines in our rebuttal experiments: (i) TwT and (ii) Retro-Search.** We did not include BREAD, as it involves reinforcement learning and optimizes a different objective, making it less directly comparable to our offline evolutionary distillation setting.
>
> The empirical results are summarized in Table R3.
>
> **TwT is a multi-teacher distillation framework** that prioritizes (1) selecting high-confidence CoTs (without explicitly checking correctness), (2) preserving sample diversity, and (3) gradually reducing reliance on explicit reasoning. This design leads to two characteristic behaviors:
> 1. TwT tends to produce more usable samples because it does not filter by correctness, and
> 2. its filtering process is coarse-grained, causing hallucinations and incorrect reasoning steps to survive more frequently.
>
> In our comparison, TwT indeed shows advantages in data utilization (e.g., improving PQA accuracy on Qwen2.5-7B), but its overall performance remains behind CoT-Evo, *largely because TwT does not enforce high-quality reasoning chains*.
>
> **Retro-Search addresses over-thinking and under-thinking by exploring the search space at each reasoning step and selecting the shortest correct trajectory.** This process helps improve CoT accuracy and correctness. In our setting, Retro-Search improves over the Single-Teacher baseline by +7.1% on ChemCoTDataset and +6.4% on BioProBench, demonstrating its strong capability. However, Retro-Search still lags behind CoT-Evo. We attribute this gap to three factors:
> 1. Retro-Search does not correct incorrect scientific reasoning inside trajectories,
> 2. it does not incorporate scientific knowledge augmentation, and
> 3. it does not encourage diverse reasoning strategies, which limits its ability to escape local optima.
>
> Across all these comparisons, **CoT-Evo demonstrates three unique strengths**:
> * It explicitly injects scientific knowledge and increases its concentration in the population via iterative evolution.
> * Its Recombination and Mutation operators enable fine-grained CoT optimization, improving both reasoning quality and diversity.
> * The Novelty-Driven Candidate Selection strategy ensures steady iterative improvement while preventing collapse to local optima.
>
> These results collectively highlight CoT-Evo’s distinct contribution beyond existing distillation and evolutionary reasoning methods.
>
> ---
>
> **Table R3. Comparison with Retro-Search and TwT**
> | **Model** | **BioProBench PQA Acc ↑** | **BioProBench ORD EM ↑** | **BioProBench ERR Acc ↑** | **BioProBench ERR F1 ↑** | **ChemCoT Und. MAE ↓** | **ChemCoT Und. TMS ↑** | **ChemCoT Und. Acc ↑** | **ChemCoT Edit Acc ↑** | **ChemCoT Reaction SR ↑** | **ChemCoT Opt. FTS ↑** |
> | ----------------------- | ------------------------- | ------------------------ | ------------------------- | ------------------------ | ---------------------- | ---------------------- | ---------------------- | ---------------------- | ------------------------- | ---------------------- |
> | Llama3.1-8B-Retro | 0.488 | 0.121 | 0.623 | 0.635 | 0.533 | 0.247 | 0.613 | 0.492 | 0.274 | 0.456 |
> | Llama3.1-8B-TwT | 0.499 | 0.397 | 0.570 | 0.459 | 0.607 | 0.238 | 0.597 | 0.506 | 0.278 | 0.433 |
> | **Llama3.1-8B-CoT-Evo** | **0.512** | **0.419** | **0.657** | **0.698** | **0.375** | **0.250** | **0.639** | **0.707** | **0.340** | **0.597** |
> | Qwen2.5-7B-Retro | 0.519 | 0.178 | 0.613 | 0.596 | 0.508 | 0.187 | 0.562 | 0.450 | 0.248 | 0.417 |
> | Qwen2.5-7B-TwT | **0.558** | 0.358 | 0.589 | 0.438 | 0.752 | 0.132 | 0.632 | 0.442 | 0.285 | 0.448 |
> | **Qwen2.5-7B-CoT-Evo** | 0.551 | **0.448** | **0.675** | **0.671** | **0.497** | **0.306** | **0.664** | **0.602** | **0.332** | **0.577** |
> | Qwen3-8B-Retro | 0.583 | 0.279 | 0.587 | 0.552 | 0.540 | 0.163 | 0.601 | 0.303 | 0.367 | 0.528 |
> | Qwen3-8B-TwT | 0.475 | 0.401 | 0.563 | 0.466 | 0.865 | 0.067 | 0.540 | 0.281 | 0.319 | 0.300 |
> | **Qwen3-8B-CoT-Evo** | **0.649** | **0.544** | **0.645** | **0.677** | **0.351** | **0.358** | **0.674** | 0.625 | 0.437 | **0.629** |

---

> ### Author Response · Authors · 2025-11-21
> **Responses to Reviewer tiKN (2)**
>
> Thank you for your valuable comment!
> > The paper does not compare its method with standard CoT under equal compute (equal token count) conditions.
> Thank you for raising this important point. We agree that comparing CoT-Evo with standard CoT-based baselines under equal computational cost is essential for isolating the true benefit of the evolutionary refinement procedure. To address this concern, we conducted an additional experiment where all methods are strictly constrained to the same total token budget during the data-generation phase.
>
> Under this equal-compute protocol, Best-of-K (BoK) and Multi-Teacher (MT) baselines operate purely by drawing more sampled trajectories until the shared budget is exhausted. Their final training CoT is selected using the same fitness function described in Section 3.2. Importantly, these baselines can only improve through larger exploration (i.e., more sampled candidates), as they lack mechanisms to revise, reorganize, or repair the reasoning content of a sampled chain.
>
> In contrast, CoT-Evo consumes the same number of tokens but distributes them differently: a portion is used for initial multi-thinker sampling (equivalent to MT/BoK), and the remaining tokens are allocated to recombination and mutation within the evolutionary loop. Once the global token budget is reached, the process halts immediately and the best trajectory in the current population is selected for training. Thus, all methods operate under strictly identical compute.
>
> The results (summarized in Table R1) show that:
>
> * **CoT-Evo consistently outperforms standard CoT, BoK, and MT under identical token budgets.** Even when the baselines can sample many more candidates, their performance grows slowly because additional exploration alone cannot repair mistakes or integrate useful reasoning fragments.
> * **CoT-Evo yields higher gains in the small- and medium-budget settings**, demonstrating that fine-grained reasoning refinement—rather than brute-force sampling—is the primary source of improvement.
> * Beyond ~50k tokens, performance continues to improve, though more gradually, indicating that CoT-Evo has not saturated and benefits from additional compute similarly to the baselines.
>
> Overall, these equal-compute experiments confirm that the improvements introduced by CoT-Evo cannot be explained by increased sampling alone. The evolutionary operations provide a qualitatively different source of benefit—systematically enhancing the structure and correctness of scientific reasoning trajectories—which persists even when compute is carefully matched. We will include these findings and further discussion in the revised manuscript.
>
> ---
>
> **Table R1. Performance under equal token-budget constraints across different models.**
> | Model               | 30K tokens             | 50K tokens            | 70K tokens           | 90K tokens            |
> | ------------------- | --------------- | --------------- | --------------- | --------------- |
> | llama3.1-8b-MT      | 0.527/0.416     | 0.529/0.442     | 0.532/0.456     | 0.531/0.462     |
> | llama3.1-8b-BoK     | 0.450/0.323     | 0.459/0.367     | 0.410/0.401     | 0.493/0.450     |
> | llama3.1-8b-CoT-Evo | 0.527/0.416 | **0.560/0.486** | **0.573/0.524** | **0.582/0.537** |
> | qwen2.5-7b-MT       | 0.516/0.430     | 0.522/0.445     | 0.528/0.461     | 0.539/0.478     |
> | qwen2.5-7b-BoK      | 0.463/0.313     | 0.475/0.347     | 0.500/0.347     | 0.516/0.360     |
> | qwen2.5-7b-CoT-Evo  | 0.516/0.430 | **0.557/0.518** | **0.584/0.565** | **0.606/0.596** |
> | qwen3-8b-MT         | 0.572/0.479     | 0.575/0.474     | 0.580/0.484     | 0.586/0.488     |
> | qwen3-8b-BoK        | 0.514/0.385     | 0.515/0.394     | 0.515/0.430     | 0.510/0.461     |
> | qwen3-8b-CoT-Evo    | 0.572/0.479 | **0.590/0.525** | **0.606/0.532** | **0.629/0.545** |
>
> ---
>
> Thank you for the comment!
> > The method is only validated on small-scale models, and it remains uncertain whether the gains would persist for larger models.
>
> We appreciate your interest in evaluating CoT-Evo on larger models. Our current experiments focus on small and medium-sized models primarily for computational efficiency and cost considerations. However, the key contribution of CoT-Evo is the generation of higher-quality CoT training data, and we expect these benefits to transfer to larger models as well. We will include additional results on larger models in the revised version. Thank you again for the helpful suggestion.

---

> ### Author Response · Authors · 2025-11-21
> **Responses to Reviewer tiKN (3)**
>
> Thank you for the careful and constructive comment!
> > Some of the reported performance improvements are not statistically significant.
>
> We acknowledge that on a small subset of tasks (most notably the PQA task), the performance differences relative to baselines are indeed modest. On PQA, all baseline methods improve over the base model by less than 0.05, and the gap between CoT-Evo and the strongest baselines also falls within a range comparable to the intrinsic noise of the task. **This reflects the high intrinsic variance and difficulty of PQA, rather than a failure of our method.**
>
> As we revisit the design of BioProBench, it is important to note that all items are derived from real laboratory protocols, and PQA in particular requires fine-grained understanding and recall of protocol details. **Without substantial domain-aligned training data or additional domain-specific prior knowledge, it is inherently challenging for a model to achieve large gains purely through reasoning.** For this class of tasks, two directions are likely to be more impactful than reasoning-only distillation: (i) enlarging the protocol-specific training corpus; (ii) incorporating mid-stage pretraining or continual pretraining on experimental-procedure–focused corpora to strengthen the model’s procedural understanding.
>
> We plan to explore these improvements in future iterations of CoT-Evo.Despite these isolated cases, we emphasize that across the majority of tasks and metrics on both scientific benchmarks, CoT-Evo exhibits consistent, stable, and sizable improvements over existing CoT distillation methods. This indicates that our evolutionary mechanism provides robust benefits for scientific reasoning overall, rather than exploiting statistical noise or overfitting to specific tasks.

---

### Official Review · Reviewer_9BYm · 2025-11-03

**Soundness:** 1
**Presentation:** 4
**Contribution:** 2
**Rating:** 2
**Confidence:** 4

**Summary:**

This paper proposes CoT-Evo, an evolutionary distillation framework for generating high-quality Chain-of-Thought (CoT) data specialized for scientific reasoning. Instead of simply selecting CoTs from one or multiple teachers, CoT-Evo iteratively evolves reasoning trajectories through an evaluation–selection–variation–update loop. It combines ideas from genetic algorithms and novelty search to maintain reasoning diversity, integrates reflective recombination and mutation operations, and uses a composite fitness function evaluating correctness, coherence, and knowledge usage. Experiments on BioProBench and ChemCoTBench demonstrate substantial gains over single-teacher, multi-teacher, and best-of-K distillation baselines, with notable improvements even against large teacher LLMs.

**Strengths:**

1. The paper is well-organized, and the pipeline diagram (Figure 1) is visually appealing and instrumental in understanding the full workflow. It clearly depicts the multi-thinker initialization, novelty-driven selection, recombination, and mutation processes, providing an intuitive grasp of the iterative framework.

2. The focus on scientific reasoning is timely and important. This area is under-explored compared with mathematical or commonsense reasoning, and the paper explicitly tackles the difficulty of distilling reasoning traces in scientific domains.

3. Experimental results show large performance improvements across all evaluated datasets compared to strong baselines (single-teacher, multi-teacher, and best-of-K). Ablations also indicate that proposed components contribute meaningfully to performance.

**Weaknesses:**

1. Although the motivation centers on scientific reasoning, the method itself is very general and could apply equally to other reasoning domains. However, the paper does not include any cross-domain experiments to validate this generality. As a result, it remains unclear why this method uniquely addresses the challenges of scientific reasoning - particularly the lack of verifiable or reliable reward signals. The intra-chain aggregation and evolutionary refinement do not appear to inject additional domain knowledge beyond what the teacher already provides.

2. The evaluation-selection-variation-update process is complex and involves multiple heuristics (novelty search, recombination triggers, mutation modes). The empirical section shows better results but does not convincingly justify why this evolutionary loop is necessary compared with simpler or alternative distillation techniques. Comparisons with other distillation baselines or under the token-budget-controlled settings would make the argument more convincing.

3. The two key points raised in the Introduction: (1) single-teacher bias and pruning insufficiency, and (2) multi-teacher diversity without fine-grained control are not unique to scientific reasoning. These limitations apply to CoT distillation in general. Therefore, the framing as “scientific reasoning-specific” seems somewhat overstated.

**Questions:**

1. Could you clarify why the proposed evolutionary process particularly benefits scientific reasoning rather than other reasoning domains? Is there empirical or theoretical evidence that the novelty-driven selection aligns better with the nature of scientific reasoning tasks?

2. Since the framework is general, it would be informative to include non-scientific benchmarks to test generalizability and demonstrate the method’s effectiveness beyond science.

3. Regarding efficiency, how does CoT-Evo’s compute cost compare to standard multi-teacher distillation or rejection-sampling pipelines? Could a smaller population or fewer iterations retain most of the gains?

4. In the fitness design, the length appropriateness and knowledge usage scores seem somewhat ad-hoc. How sensitive is the final performance to these weighting hyperparameters (λ₁, λ₂)?

---

> ### Author Response · Authors · 2025-11-21
> **Responses to Reviewer 9BYm (1)**
>
> Thank you very much for your valuable comments!
> > Although the motivation centers on scientific reasoning, the method itself is very general and could apply equally to other reasoning domains. However, the paper does not include any cross-domain experiments to validate this generality. As a result, it remains unclear why this method uniquely addresses the challenges of scientific reasoning - particularly the lack of verifiable or reliable reward signals. The intra-chain aggregation and evolutionary refinement do not appear to inject additional domain knowledge beyond what the teacher already provides.
>
> > Could you clarify why the proposed evolutionary process particularly benefits scientific reasoning rather than other reasoning domains? Is there empirical or theoretical evidence that the novelty-driven selection aligns better with the nature of scientific reasoning tasks?
>
> > Since the framework is general, it would be informative to include non-scientific benchmarks to test generalizability and demonstrate the method’s effectiveness beyond science.
>
> We appreciate your recognition of our method’s strong performance on scientific tasks in the Strengths section.
>
> **Q1: Why we focus specifically on scientific reasoning?**
>
> **Our primary objective is to improve LLM reasoning in professional scientific domains**, and our method is intentionally designed for scientific reasoning scenarios. In highly specialized tasks, such as experimental protocol design or molecular editing, although short-format QA datasets exist, there is an acute lack of high-quality long CoT data. Therefore, an effective CoT distillation paradigm must (i) extract as many reasoning trajectories as possible from limited data, and (ii) refine these trajectories at a fine-grained level to ensure both scientific correctness and final-answer accuracy. Our experiments show that existing distillation approaches remain limited in data efficiency and reasoning quality. By incorporating scientific knowledge and intra-chain aggregation, **CoT-Evo continually evolves CoTs toward higher accuracy and diversity**, producing more and higher-quality scientific reasoning traces.
>
> **Q2: Why CoT-Evo’s design particularly benefits scientific reasoning?**
>
> Each module of CoT-Evo is deliberately aligned with the characteristics of scientific reasoning:
> 1. **Knowledge Augmentation in CoT Initialization**. Even advanced LLMs (e.g., DeepSeek-R1) exhibit knowledge-usage hallucination during complex scientific reasoning, leading to incorrect chains (see Appendix, Case 1, line 1260). We explicitly inject verified scientific knowledge during initialization and increase its "concentration" through subsequent evolutionary steps. We believe this step becomes even more crucial in more specialized domains, and future work may incorporate retrieval-augmented strategies for further improvement.
> 2. **Knowledge-Usage Correctness as part of the fitness function**. This reward is not meant to introduce additional domain knowledge beyond the teacher; rather, it ensures that candidate CoTs evolve toward more accurate use of the knowledge already required by the task. Empirically, the final candidate pool contains CoTs with more accurate scientific knowledge usage (Table 2, line 410). Importantly, the reward signals we use are verifiable and reliable: Exact Match and Length Appropriateness are fully deterministic, while Knowledge-Usage Correctness, though LLM-judged, is grounded by providing a reference knowledge set $K_x$, which makes the evaluation stable and reliable.
> 3. **Novelty-driven selection (NSLC-based) for exploration**. At each iteration, we adopt a novelty-driven selection mechanism based on NSLC (a well-validated algorithm in evolutionary search). This ensures that CoT-Evo preserves not only the more accurate trajectories but also those with innovative and diverse reasoning patterns, enriching the final pool of scientific CoTs (Figure 2, line 378).
>
> **Q3: On generality beyond scientific reasoning.**
>
> Although our design is motivated by scientific reasoning challenges, we agree that the framework is general and can be adapted to other domains. We view cross-domain evaluation as supplementary validation, not as a prerequisite for the method's scientific relevance. To address your suggestion, we are already running CoT-Evo on additional non-scientific reasoning benchmarks (e.g., mathematical reasoning), and we will report the corresponding results within the next few days.

---

> > ### Author Response · Authors · 2025-11-21
> > **Responses to Reviewer 9BYm (3)**
> >
> > Thank you for the insightful comment.
> > > The two key points raised in the Introduction: (1) single-teacher bias and pruning insufficiency, and (2) multi-teacher diversity without fine-grained control are not unique to scientific reasoning. These limitations apply to CoT distillation in general. Therefore, the framing as “scientific reasoning-specific” seems somewhat overstated.
> >
> > We sincerely appreciate the reviewer’s observation. We fully agree that, at a phenomenological level, these two limitations are not exclusive to scientific reasoning and indeed appear broadly in CoT distillation. In the revision, we will soften the phrasing that suggests these issues are “scientific-reasoning-specific.”
> > At the same time, we would like to clarify the intended framing: our goal is not to claim the discovery of new failure modes that exist only in scientific domains. Instead, we aim to highlight that **these general CoT distillation challenges become significantly amplified in scientific reasoning**, due to the unique structure and data characteristics of scientific tasks. As a result, their negative impact on model performance is more severe than in typical mathematical or commonsense reasoning.
> > Concretely, scientific reasoning differs from general reasoning tasks in three key ways:
> >
> > 1. **Higher and more subtle teacher error rates.** Even state-of-the-art reasoning LLMs frequently produce long scientific CoTs that appear superficially plausible but contain mechanistic inaccuracies or incomplete domain knowledge. Under such conditions, single-teacher distillation and token-level pruning often “compress” an already flawed chain rather than correcting it; worse, pruning may remove explanatory segments and make domain-knowledge hallucination harder to detect.
> > 2. **More structured reasoning chains where mechanistic correctness matters more than the final answer.** In benchmarks such as ChemCoTBench and BioProBench, models must not only output the correct final decision (e.g., molecule satisfies constraints, experiment step ordering is valid) but also provide mechanistically sound rationales involving molecular edits, experimental conditions, and causal dependencies. Multi-teacher BoK therefore reduces to selecting one complex chain that “looks best,” but without fine-grained intra-chain recombination and error correction, the distilled student may still inherit incorrect or incomplete mechanistic knowledge even when the final answer happens to be correct.
> > 3. **Significantly sparser reward signals.** Unlike mathematics or code, scientific reasoning lacks step-level executable supervision; typically only the final result and a few structural constraints are verifiable (e.g., property thresholds, dependency checks). Thus, fine-grained control over knowledge usage and logical structure becomes essential. Our composite fitness function, final answer correctness + length priors derived from scientific datasets + knowledge-usage correctness relative to reference fragments $K_x$, is designed precisely to operate under this sparse-reward scientific environment.
> >
> > Given these characteristics, the two nominally “general” limitations are far more consequential in scientific reasoning:
> >
> > * Single-teacher bias & pruning insufficiency do not merely affect reasoning style or verbosity (as in general benchmarks) but directly propagate incorrect or one-sided scientific knowledge, which materially harms the student model’s understanding of scientific rules and experimental protocols.
> > * Multi-teacher diversity without fine-grained control is especially critical in knowledge-intensive scientific scenarios. For example, in experimental-procedure design, different trajectories often contain complementary mechanistic details. Without intra-chain recombination and repair, coarse chain-level selection fails to realize the benefits of multi-teacher supervision.
> >
> > In the revision, we will make this distinction clearer by rephrasing our discussion from “scientific reasoning-specific limitations” to “limitations that are particularly pronounced and consequential in scientific reasoning settings”, which more accurately reflects the intended contribution and positioning of the paper.

---

> ### Author Response · Authors · 2025-11-21
> **Responses to Reviewer 9BYm (2)**
>
> Thank you for the insightful comments!
>
> > The evaluation-selection-variation-update process is complex and involves multiple heuristics (novelty search, recombination triggers, mutation modes). The empirical section shows better results but does not convincingly justify why this evolutionary loop is necessary compared with simpler or alternative distillation techniques. Comparisons with other distillation baselines or under the token-budget-controlled settings would make the argument more convincing.
>
> > Regarding efficiency, how does CoT-Evo’s compute cost compare to standard multi-teacher distillation or rejection-sampling pipelines? Could a smaller population or fewer iterations retain most of the gains?
>
> We fully agree with your concern. CoT-Evo indeed introduces an iterative evolutionary loop, and this naturally consumes more tokens than a single-pass sampling strategy. To verify that the performance gains do not simply stem from increased token usage, we conducted additional experiments where all methods, Best-of-K (BoK), Multi-Teacher (MT), and CoT-Evo, were constrained to the same total distillation token budget.
>
> Concretely, for BoK, we repeatedly sample CoTs from a single teacher until the token budget is reached and then select the best CoT using the same fitness function described in Section 3.2; we report the best result across all teachers. For MT, we sequentially distill from six teachers; after each full round (one sample per teacher), if the token budget is not reached, the process continues into the next round. For CoT-Evo, we monitor the token usage of the entire pipeline and terminate the evolutionary loop immediately when the budget is reached, selecting the best candidate from the current population. The results are shown in Table R1.
>
> These controlled-budget experiments lead to two major observations:
> 1. **First, the evolutionary loop is indeed necessary**: although MT and BoK benefit moderately from increased sampling, their gains scale slowly with additional tokens. In contrast, CoT-Evo substantially improves CoT quality through fine-grained recombination and mutation, enabling the student model to achieve markedly higher performance even under identical token constraints.
> 2. **Second, CoT-Evo exhibits fast improvement under small token budgets and slows only after ~50K tokens.** We observe this both in Table R1 and in Figure 3. This suggests that with limited population, iterations, and token budget, the evolutionary loop has not saturated—performance continues to improve as more budget becomes available. We therefore expect further gains with larger budgets.
>
> In summary, the token-controlled experiments directly validate that CoT-Evo’s gains cannot be attributed to increased sampling alone. Instead, the evolutionary loop provides a principled mechanism for iterative CoT refinement, enabling the student model to benefit from higher-quality reasoning trajectories even under identical computational budgets. We will include these new results and clarifications in the revised submission.
>
> **Table R1. Performance under equal token-budget constraints across different models.**
> | Model               | 30K tokens             | 50K tokens            | 70K tokens           | 90K tokens            |
> | ------------------- | --------------- | --------------- | --------------- | --------------- |
> | llama3.1-8b-MT      | 0.527/0.416     | 0.529/0.442     | 0.532/0.456     | 0.531/0.462     |
> | llama3.1-8b-BoK     | 0.450/0.323     | 0.459/0.367     | 0.410/0.401     | 0.493/0.450     |
> | llama3.1-8b-CoT-Evo | 0.527/0.416 | **0.560/0.486** | **0.573/0.524** | **0.582/0.537** |
> | qwen2.5-7b-MT       | 0.516/0.430     | 0.522/0.445     | 0.528/0.461     | 0.539/0.478     |
> | qwen2.5-7b-BoK      | 0.463/0.313     | 0.475/0.347     | 0.500/0.347     | 0.516/0.360     |
> | qwen2.5-7b-CoT-Evo  | 0.516/0.430 | **0.557/0.518** | **0.584/0.565** | **0.606/0.596** |
> | qwen3-8b-MT         | 0.572/0.479     | 0.575/0.474     | 0.580/0.484     | 0.586/0.488     |
> | qwen3-8b-BoK        | 0.514/0.385     | 0.515/0.394     | 0.515/0.430     | 0.510/0.461     |
> | qwen3-8b-CoT-Evo    | 0.572/0.479 | **0.590/0.525** | **0.606/0.532** | **0.629/0.545** |

---

> ### Author Response · Authors · 2025-11-21
> **Responses to Reviewer 9BYm (4)**
>
> Thank you for your helpful comment!
> > In the fitness design, the length appropriateness and knowledge usage scores seem somewhat ad-hoc. How sensitive is the final performance to these weighting hyperparameters (λ₁, λ₂)?
>
> We apologize for not explaining the choice of the fitness weights $\lambda_1$ and $\lambda_2$ more clearly in the main text. In fact, we conducted a small sensitivity study in the early stage of our experiments using **llama3.1-8b-instruct** as the student model. We varied $\lambda_1 \in {0.1, 0.3, 0.5}$ and $\lambda_2 \in {0.1, 0.2, 0.3}$, and report the averaged results on BioProBench and ChemCoTBench in Table R2 below.
>
> Empirically, we find that the final performance is **not very sensitive** to $\lambda_1$ within the tested range. This is consistent with the fact that most evolved trajectories already fall into a reasonable length range, so the contribution of $s_{\text{LEN}}$ mainly acts as a mild regularizer rather than a dominant term. Similarly, when $\lambda_2 \in {0.1, 0.2}$, we observe no substantial performance differences: in this regime, the weight on knowledge-usage correctness is already large enough to filter out obviously bad trajectories, so even $\lambda_2 = 0.1$ suffices to obtain high-quality CoTs. However, when $\lambda_2$ is further increased to 0.3, we see a degradation on several tasks. Our analysis is that increasing $\lambda_2$ too much effectively down-weights the exact-match term $s_{\text{EM}}$, which leads to an over-selection of trajectories that use the knowledge in a seemingly correct way but still produce incorrect final answers. Based on this trade-off, we choose $\lambda_1 = 0.3, \lambda_2 = 0.1$ as a robust default that balances answer correctness, length regularization, and knowledge usage. We will add this explanation and the following table to the revised version to improve clarity and reproducibility.
>
> **Table R2. Sensitivity of CoT-Evo to $\lambda_1, \lambda_2$ on llama3.1-8b-instruct**
> | $\lambda_1, \lambda_2$ | BioProBench PQA Acc ↑ | BioProBench ORD EM ↑ | BioProBench ERR Acc ↑ | BioProBench ERR F1 ↑ | ChemCoT Und. MAE ↓ | ChemCoT Und. TMS ↑ | ChemCoT Und. Acc ↑ | ChemCoT Edit Acc ↑ | ChemCoT Reaction SR ↑ | ChemCoT Opt. FTS ↑ |
> | ---------------------- | ------------------------- | ------------------------ | ------------------------- | ------------------------ | ---------------------- | ---------------------- | ---------------------- | ---------------------- | ------------------------- | ---------------------- |
> | 0.1, 0.1 | 0.471 | 0.408 | 0.624 | 0.679 | 0.415 | 0.281 | **0.680** | 0.618 | 0.311 | 0.591 |
> | **0.3, 0.1 (default)** | 0.512 | **0.419** | 0.657 | **0.698** | 0.375 | 0.250 | 0.639 | **0.707** | 0.340 | **0.597** |
> | 0.5, 0.1 | 0.514 | 0.415 | 0.653 | 0.694 | 0.480 | 0.268 | 0.619 | 0.656 | **0.368** | 0.586 |
> | 0.3, 0.2 | **0.525** | 0.410 | **0.681** | 0.668 | 0.417 | **0.257** | 0.607 | 0.680 | 0.349 | 0.567 |
> | 0.3, 0.3 | 0.484 | 0.398 | 0.533 | 0.678 | **0.287** | 0.168 | 0.519 | 0.656 | 0.268 | 0.539 |

---

### Author Response · Authors · 2025-11-30
**Response Summary to Area Chair**

We appreciate the opportunity to address the reviewers' insightful comments during the discussion phase. We have carefully considered and resolved all major concerns raised, providing additional experiments, clarifications, and refinements. Below, we summarize our responses to four key points that were highlighted by the reviewers:
### 1. Why is the proposed evolutionary process specifically designed for scientific reasoning?
Several reviewers, including Reviewer 9BYm and 7Nvu, raised this question. We clarify that our primary objective is to improve LLM performance in specialized scientific domains where **there is a critical need for high-quality long CoT data**, a gap that existing methods fail to address adequately. CoT-Evo is intentionally designed to **refine reasoning traces in scientific tasks** where correctness and mechanistic accuracy are paramount, such as experimental protocol design or molecular editing. Our approach enhances **knowledge usage and intra-chain aggregation**, allowing for the evolution of CoTs that better capture scientific reasoning patterns. Although our method is adaptable to other domains, the focus on scientific reasoning stems from the unique challenges of these tasks.
### 2. CoT-Evo’s advantage under token-budget-controlled settings
Reviewers 9BYm, pgWS, and 7Nvu raised concerns about whether CoT-Evo offers advantages over simple baselines, given the iterative nature of its evolutionary loop. In response, we conducted controlled experiments where all methods (Best-of-K, Multi-Teacher, and CoT-Evo) were constrained to the **same total token budget**. The results, summarized in Table R1, show that CoT-Evo’s **evolutionary loop enables substantial improvements in CoT quality** even under identical token constraints, outperforming baseline methods that primarily rely on sampling. Our findings indicate that while the other methods benefit from increased sampling, CoT-Evo refines the reasoning process, leading to significant gains in reasoning quality.
### 3. Hyperparameter sensitivity
Reviewers 9BYm and pgWS highlighted the sensitivity of the evolutionary process to hyperparameters, particularly fitness weights and the k-nearest neighbors parameter. We conducted further analysis on the **sensitivity of key hyperparameters such as $k$, and the fitness weights $\lambda_1$ and $\lambda_2$.** For the novelty search parameter $k$, Table R4 shows that $k=2$ yields the best relative performance. This setting effectively balances stability with discriminativeness, avoiding the noise sensitivity of $k=1$ while preventing the over-smoothing of local structures seen with larger values. Regarding fitness weights, Table R2 demonstrates that the system is robust as long as the exact-match signal remains the dominant objective. Our chosen configuration ($\lambda_1=0.3, \lambda_2=0.1$) represents a stable optimum; distinctively, our analysis reveals that excessively increasing $\lambda_2$ degrades performance by incentivizing knowledge retrieval at the expense of final reasoning accuracy.
### 4. Generalizability of CoT-Evo to non-scientific domains
Finally, Reviewer 9BYm and 7Nvu raised concerns about the generalizability of CoT-Evo beyond scientific reasoning tasks. While we acknowledge that our method was designed with scientific reasoning as the primary focus, we are pleased that reviewers recognize its potential universality. To address this, we validated our method's potential on general mathematical and STEM tasks **using the LIMO dataset (800 questions) as the initial data pool**. As shown in Table R5, CoT-Evo achieved the highest average performance (64.21%) on Qwen2.5-7B by generating superior CoTs, highlighting the primacy of data quality. Notably, methods like Single Teacher (ST) and Multi-Teacher (MT) suffered due to low data utilization rates (40.75% and 59.87%), while the LIMO baseline, despite high utilization (100%), was marginally outperformed by CoT-Evo (95.75% utilization), confirming the benefits of our evolutionary fine-tuning beyond the scientific domain.

Additionally, we included recent baselines, TwT and Retro-Search, and our results in Table R3 show that CoT-Evo outperforms them by leveraging fine-grained recombination and mutation for iterative CoT refinement.

The related experiments have been included in the revised version of the paper. We believe that the revisions we are making will clarify and strengthen our paper’s contributions, addressing all the concerns raised during the discussion phase. We are confident that these improvements will make the case for CoT-Evo’s **novelty**, **effectiveness**, and **practical value** even more compelling.

---

> ### Author Response · Authors · 2025-11-30
>
> **Table R1. Performance under equal token-budget constraints across different models.**
> | Model               | 30K tokens             | 50K tokens            | 70K tokens           | 90K tokens            |
> | ------------------- | --------------- | --------------- | --------------- | --------------- |
> | llama3.1-8b-MT      | 0.527/0.416     | 0.529/0.442     | 0.532/0.456     | 0.531/0.462     |
> | llama3.1-8b-BoK     | 0.450/0.323     | 0.459/0.367     | 0.410/0.401     | 0.493/0.450     |
> | llama3.1-8b-CoT-Evo | 0.527/0.416 | **0.560/0.486** | **0.573/0.524** | **0.582/0.537** |
> | qwen2.5-7b-MT       | 0.516/0.430     | 0.522/0.445     | 0.528/0.461     | 0.539/0.478     |
> | qwen2.5-7b-BoK      | 0.463/0.313     | 0.475/0.347     | 0.500/0.347     | 0.516/0.360     |
> | qwen2.5-7b-CoT-Evo  | 0.516/0.430 | **0.557/0.518** | **0.584/0.565** | **0.606/0.596** |
> | qwen3-8b-MT         | 0.572/0.479     | 0.575/0.474     | 0.580/0.484     | 0.586/0.488     |
> | qwen3-8b-BoK        | 0.514/0.385     | 0.515/0.394     | 0.515/0.430     | 0.510/0.461     |
> | qwen3-8b-CoT-Evo    | 0.572/0.479 | **0.590/0.525** | **0.606/0.532** | **0.629/0.545** |
>
> ---
>
> **Table R2. Sensitivity of CoT-Evo to $(\lambda_1, \lambda_2)$ on llama3.1-8b-instruct**
>
> | $(\lambda_1, \lambda_2)$ | **BioProBench PQA** Acc ↑ | **BioProBench ORD** EM ↑ | **BioProBench ERR** Acc ↑ | **BioProBench ERR** F1 ↑ | **ChemCoT Und.** MAE ↓ | **ChemCoT Und.** TMS ↑ | **ChemCoT Und.** Acc ↑ | **ChemCoT Edit** Acc ↑ | **ChemCoT Reaction** SR ↑ | **ChemCoT Opt.** FTS ↑ |
> | --| -- | -- | -- | -- | -- | -- | -- | -- | -- | -- |
> | 0.1, 0.1 | 0.471 | 0.408 | 0.624 | 0.679 | 0.415 | **0.281** | **0.680** | 0.618 | 0.311 | 0.591 |
> | **0.3, 0.1 (default)** | 0.512 | **0.419** | 0.657 | **0.698** | 0.375 | 0.250 | 0.639 | **0.707** | 0.340 | **0.597** |
> | 0.5, 0.1 | 0.514 | 0.415 | 0.653 | 0.694 | 0.480 | 0.268 | 0.619 | 0.656 | **0.368** | 0.586 |
> | 0.3, 0.2 | **0.525** | 0.410 | **0.681** | 0.668 | 0.417 | 0.257 | 0.607 | 0.680 | 0.349 | 0.567 |
> | 0.3, 0.3 | 0.484 | 0.398 | 0.533 | 0.678 | **0.287** | 0.168 | 0.519 | 0.656 | 0.268 | 0.539 |
>
> ---
>
> **Table R3. Comparison with Retro-Search and TwT**
> | **Model** | **BioProBench PQA Acc ↑** | **BioProBench ORD EM ↑** | **BioProBench ERR Acc ↑** | **BioProBench ERR F1 ↑** | **ChemCoT Und. MAE ↓** | **ChemCoT Und. TMS ↑** | **ChemCoT Und. Acc ↑** | **ChemCoT Edit Acc ↑** | **ChemCoT Reaction SR ↑** | **ChemCoT Opt. FTS ↑** |
> | ----------------------- | ------------------------- | ------------------------ | ------------------------- | ------------------------ | ---------------------- | ---------------------- | ---------------------- | ---------------------- | ------------------------- | ---------------------- |
> | Llama3.1-8B-Retro | 0.488 | 0.121 | 0.623 | 0.635 | 0.533 | 0.247 | 0.613 | 0.492 | 0.274 | 0.456 |
> | Llama3.1-8B-TwT | 0.499 | 0.397 | 0.570 | 0.459 | 0.607 | 0.238 | 0.597 | 0.506 | 0.278 | 0.433 |
> | **Llama3.1-8B-CoT-Evo** | **0.512** | **0.419** | **0.657** | **0.698** | **0.375** | **0.250** | **0.639** | **0.707** | **0.340** | **0.597** |
> | Qwen2.5-7B-Retro | 0.519 | 0.178 | 0.613 | 0.596 | 0.508 | 0.187 | 0.562 | 0.450 | 0.248 | 0.417 |
> | Qwen2.5-7B-TwT | **0.558** | 0.358 | 0.589 | 0.438 | 0.752 | 0.132 | 0.632 | 0.442 | 0.285 | 0.448 |
> | **Qwen2.5-7B-CoT-Evo** | 0.551 | **0.448** | **0.675** | **0.671** | **0.497** | **0.306** | **0.664** | **0.602** | **0.332** | **0.577** |
> | Qwen3-8B-Retro | 0.583 | 0.279 | 0.587 | 0.552 | 0.540 | 0.163 | 0.601 | 0.303 | 0.367 | 0.528 |
> | Qwen3-8B-TwT | 0.475 | 0.401 | 0.563 | 0.466 | 0.865 | 0.067 | 0.540 | 0.281 | 0.319 | 0.300 |
> | **Qwen3-8B-CoT-Evo** | **0.649** | **0.544** | **0.645** | **0.677** | **0.351** | **0.358** | **0.674** | 0.625 | 0.437 | **0.629** |
>
>
> ---
> **Table R4. Sensitivity of CoT-Evo to $k$ (k-nearest neighbors) on llama3.1-8b-instruct**
>
> | (k) | **BioProBench PQA** Acc ↑ | **BioProBench ORD** EM ↑ | **BioProBench ERR** Acc ↑ | **BioProBench ERR** F1 ↑ | **ChemCoT Und.** MAE ↓ | **ChemCoT Und.** TMS ↑ | **ChemCoT Und.** Acc ↑ | **ChemCoT Edit** Acc ↑ | **ChemCoT Reaction** SR ↑ | **ChemCoT Opt.** FTS ↑ |
> | -- | -- | ------------------------ | ------------------------- | ------------------------ | ---------------------- | ---------------------- | ---------------------- | ---------------------- | ------------------------- | ---------------------- |
> | 1 | 0.508 | 0.395 | 0.638 | 0.681 | 0.427 | 0.230 | 0.631 | 0.653 | 0.339 | 0.567 |
> | **2 (default)** | **0.512** | 0.419 | **0.657** | **0.698** | **0.375** | 0.250 | 0.639 | **0.707** | 0.340 | **0.597** |
> | 3 | 0.502 | **0.426** | 0.648 | 0.693 | 0.405 | **0.348** | **0.669** | 0.684 | **0.348** | 0.595 |

---

> ### Author Response · Authors · 2025-11-30
>
> **Table R5: Comparison of model performance (pass@1) across various general reasoning benchmarks.**
> | Benchmarks        | Base Model (qwen2.5-7b-instruct) | ST    | MT    | BoK   | LIMO  | CoT-Evo |
> |---------------|---------|-------|-------|-------|-------|---------|
> | AIME24        | 13.33   | 23.08 | 20.00 | 25.00 | 10.00 | **45.00** |
> | AMC23         | 47.50   | 70.59 | 57.89 | 75.00 | 75.00 | **85.71** |
> | MATH500       | 75.40   | 80.11 | 86.67 | 89.20 | **91.80** | 88.80 |
> | OlympiadBench | 36.89   | 52.17 | 64.66 | 67.87 | **70.75** | 66.14 |
> | CHMath        | 13.33   | 37.50 | 26.67 | 26.67 | 43.33 | **50.00** |
> | Gaokao        | 29.11   | 59.15 | 70.27 | 68.97 | 72.15 | **73.41** |
> | Kaoyan        | 30.15   | 41.14 | 44.81 | 40.74 | 46.10 | **47.24** |
> | GradeSchool   | 40.48   | 49.46 | 54.01 | **71.31** | 59.21 | 60.00 |
> | Minerva       | 49.63   | 48.36 | 49.46 | 51.03 | 58.22 | **60.54** |
> | GPQA          | 47.47   | 40.21 | 40.96 | 42.86 | 50.00 | **52.87** |
> | AVG.          | 38.33   | 50.18 | 51.73 | 59.64 | 62.19 | **64.21** |

---

### Meta-Review · Area_Chair_riDC · 2026-01-07

**Summary:**

The reviewers raised several concerns about CoT-Evo, focusing on its computational efficiency, sensitivity to hyperparameters, generalizability beyond scientific reasoning, and comparisons with other methods. Specifically, they questioned whether the iterative nature of CoT-Evo justified its higher compute cost and whether the method's performance was overly sensitive to hyperparameters. There were also concerns about the lack of experiments in non-scientific domains, and the paper's failure to compare CoT-Evo with recent distillation or evolutionary methods. Additionally, some performance improvements were not statistically significant. The authors addressed these concerns by providing additional experiments showing CoT-Evo's superior performance under equal token budgets, demonstrating its robustness to hyperparameter changes, and expanding the evaluation to non-scientific tasks, while also clarifying its advantages over competing methods.

**Reviewer Concerns:**

The authors effectively addressed several reviewer concerns in their rebuttal. They clarified the computational efficiency of CoT-Evo by showing that it outperforms baseline methods under equal token budgets, resolving doubts about its additional compute cost. They also provided a detailed sensitivity analysis for key hyperparameters, demonstrating that CoT-Evo is relatively robust to variations in these settings. Additionally, they responded to concerns about generalizability by including non-scientific domain evaluations, strengthening their argument for CoT-Evo’s broader applicability.

However, some concerns remain unresolved. Despite the additional comparisons, reviewers still question whether CoT-Evo’s performance improvements are statistically significant, particularly for tasks like PQA where improvements were modest. Furthermore, while the authors demonstrated CoT-Evo’s effectiveness on small models, there is still uncertainty regarding whether the method would scale effectively to larger models, a point that was not fully addressed in the rebuttal.

**Reviewer Scores:**

eviewer 9BYm: This reviewer initially rated the paper as a "reject" (score of 2), expressing concerns about the generality of CoT-Evo and its evolutionary loop’s necessity. After the authors addressed these concerns with additional experiments on non-scientific domains and demonstrated the method’s advantages under equal token budgets, it's likely that the reviewer would have been more convinced by the performance improvements and the clarity provided on CoT-Evo’s scientific focus. The reviewer's score could have improved to a "marginal accept" (score of 4), acknowledging the novelty and strengths of the method while remaining cautious about its scalability and the generality of its benefits.

Reviewer tiKN: This reviewer gave a score of 6, noting some doubts about the computational cost and the lack of comparison to recent methods. With the additional comparisons and the detailed rebuttal regarding CoT-Evo’s efficiency under equal compute conditions, the reviewer might have slightly increased their score, recognizing the distinctiveness of the method. However, their concerns about statistical significance and the method's performance on larger models were not fully addressed, so their score would likely remain a "marginal accept" (score of 6), but with a higher confidence in the novelty and robustness of the approach.

Reviewer pgWS: This reviewer also rated the paper as a score of 6, highlighting the computational overhead and sensitivity to hyperparameters. The authors’ rebuttal provided key clarifications and additional experiments, especially regarding the efficiency and sensitivity of the method to various parameters. However, the concern about the potential flaws in the LLM-based evaluation process and the still-uncertain scalability to larger models might have kept the reviewer from fully embracing the method. The score could have remained at "marginal accept" (score of 6), but with greater confidence in the paper’s contribution to evolutionary reasoning methods.

Reviewer 7Nvu: This reviewer initially gave a score of 6, noting the lack of evaluation on other reasoning domains and additional benchmarks. After the authors expanded the evaluation to non-scientific tasks and clarified the co-evolution process, the reviewer would likely have increased their score to "accept" (7), appreciating the broader applicability and deeper insights into the iterative refinement process.

---

### Decision · Program_Chairs · 2026-01-26

Accept (Poster)